# 🦙 IWR-Bench: Can LVLMs reconstruct interactive webpage from a user interaction video?

**Yang Chen**[*,1,2], **Minghao Liu**[*,3,4], **Yufan Shen**[*,†,1], **Yunwen Li**[*,4,5], **Tianyuan Huang**[*,2],
**Xinyu Fang**[*,1,2], **Tianyu Zheng**[4], **Wenxuan Huang**[6], **Cheng Yang**[1], **Daocheng Fu**[1,7],
**Jianbiao Mei**[1,2], **Rong Wu**[1,2], **Licheng Wen**[1], **Xuemeng Yang**[1], **Song Mao**[1], **Qunshu Lin**[3],
**Zhi Yu**[2], **Yongliang Shen**[2], **Yu Qiao**[1], **Botian Shi**[†,1,8]

[1]Shanghai Artificial Intelligence Laboratory, [2]Zhejiang University, [3]2077AI, [4]M-A-P,
[5]Chinese University of Hong Kong (Shenzhen), [6]Chinese University of Hong Kong,
[7]Fudan University, [8]Shanghai Innovation Institute
shenyfzju@gmail.com, shibotian@pjlab.org.cn

## Abstract

The webpage-to-code task requires models to understand visual representations of webpages and generate corresponding code. However, existing benchmarks primarily focus on static screenshot-to-code tasks, thereby overlooking the dynamic interactions fundamental to real-world web applications. To address this limitation, this paper introduces IWR-Bench, a novel benchmark for evaluating the capabilities of Large Vision-Language Models (LVLMs) in interactive webpage reconstruction from video. IWR-Bench comprises 113 meticulously curated tasks from 100 real-world websites, with 1,001 actions and featuring diverse interaction complexities (e.g., web games), visual styles, and domains. Aligning with standard web development practices, each task includes not only user interaction videos but also all crawled static assets (e.g., images, videos). This benchmark evaluates models on two fundamental challenges: comprehensive multi-modal reasoning to infer interaction logic from video and assets, and advanced code generation to translate this logic into functional code. An agent-as-a-judge framework with a comprehensive metric system automatically assesses the functional correctness and visual fidelity of generated webpages. Extensive experiments on 28 LVLMs reveal a significant challenge: the best model achieves an overall score of only 36.35%, as functional correctness (24.39% IFS) lags significantly behind visual fidelity (64.25% VFS). These results highlight critical limitations in current models' ability to reason about temporal dynamics and synthesize event-driven logic, establishing IWR-Bench as a challenging frontier for vision-language research. The benchmark and evaluation code would be made publicly available.

## 1 Introduction

Recent advances in Large Vision-Language Models (LVLMs) have unlocked remarkable capabilities in visual understanding and code generation (OpenAI, 2025; Comanici et al., 2025; Bai et al., 2025). State-of-the-art models can now translate a static screenshot of a webpage into corresponding HTML with impressive fidelity (Yun et al., 2024; Gui et al., 2025). This nascent success, however, highlights a fundamental limitation of current evaluation methodologies. Existing benchmarks are either confined to static reconstruction (e.g., Design2Code (Si et al., 2024), WebSight (Laurençon et al., 2024)) or model interactions as single-step, stateless events from image pairs (e.g., Interaction2Code (Xiao et al., 2025)), while also failing to provide the necessary static assets for reconstruction. This simplified setup falls short of capturing the continuous, stateful workflows and complete resource context characteristic of real-world web applications. The disconnect between demonstrated capabilities and the demands of true interactivity motivates our central research question: **Can LVLMs reconstruct the dynamic, interactive functionalities of a webpage from observing a user interaction video?**

---

\* Equal contribution, † Corresponding authors.

Reconstructing an interactive webpage from video poses two fundamental challenges. The first, **comprehensive multi-modal perception and reasoning** (Luo et al., 2024; Gupta & Kembhavi, 2023; Song et al., 2025; Deka et al., 2017; Lee et al., 2023), is the process of inferring latent interaction logic from dynamic visual evidence. This requires a model to ground its temporal understanding of observed interactions in a precise visual comprehension of the resultant UI states. A critical facet of this reasoning is robust image matching to associate dynamic elements with their static asset counterparts. The second challenge, **advanced code generation** (Jimenez et al., 2024; Xiao et al., 2025; Li et al., 2022), is the translation of this inferred logic into functional code that implements the complex, stateful logic of interactive applications (e.g., web-games like 2048 and Minesweeper).

The construction of a comprehensive benchmark for interactive webpage reconstruction confronts three pivotal challenges. The first pertains to ensuring **Diverse Interaction Coverage**, which necessitates the curation of tasks spanning a broad spectrum of interaction paradigms and visual complexities, while simultaneously adhering to strict standardization for reproducible evaluation. The second challenge centers on the establishment of an **Authentic Task Environment**. Departing from prior benchmarks characterized by incomplete setups or placeholder assets (Jiang et al., 2025; Gui et al., 2025), this requires the meticulous curation of a complete set of authentic resources from live websites. Such resources must encompass both static assets, such as images and icons, and dynamic content, such as embedded videos, to faithfully represent real-world development contexts. The final challenge lies in the formulation of a **Robust Automated Evaluation** protocol. Conventional metrics, including pixel-wise similarity, are insufficient for this purpose (Zhang et al., 2018; Caron et al., 2021; Radford et al., 2021), as they cannot appraise functionality. An effective protocol must therefore employ programmatic interaction with the generated webpage to ascertain both the functional integrity of its components and the state-wise visual consistency across dynamic transitions.

This paper formalizes the task of Interactive Webpage Reconstruction (IWR) and introduces IWR-Bench, a comprehensive benchmark that addresses these fundamental design challenges. To ensure comprehensive coverage, tasks are taxonomized along orthogonal axes of application domain, visual complexity, and interaction logic, as illustrated in Figure 3. Each task instance, as depicted in Figure 2, then provides the model with (a) an interaction video that captures a complete, stateful workflow, and (b) the full set of crawled static assets. This setup ensures a realistic reconstruction context. Evaluation is conducted via programmatic interaction: an 'agent-as-a-judge' executes a ground-truth (c) action sequence to assess the generated webpage's functionality. Performance is quantified by two holistic metrics: the Interactive Functionality Score (IFS), a unified measure of operational and logical correctness, and the Visual Fidelity Score (VFS), a composite metric integrating low-level features with high-level semantic evaluation.

An extensive evaluation on 28 leading LVLMs reveals substantial challenges posed by the IWR task. The top-performing proprietary model, GPT-5, achieves a Final Score of 36.35%. A clear performance hierarchy is observed, as leading open-source models attain lower scores, and video-specialized models lag even further behind. For the top-performing model, a significant disparity exists between its functional correctness (24.39% IFS) and visual fidelity (64.25% VFS). This gap indicates a fundamental limitation across the field: while models can reproduce static layouts with

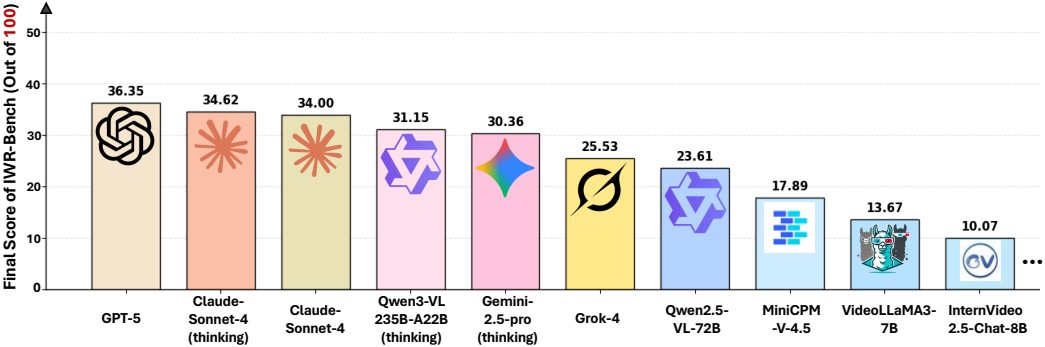

Figure 1: Performance of 10 representative models on IWR-Bench. For a comprehensive list of all 28 model results, see Table 3.

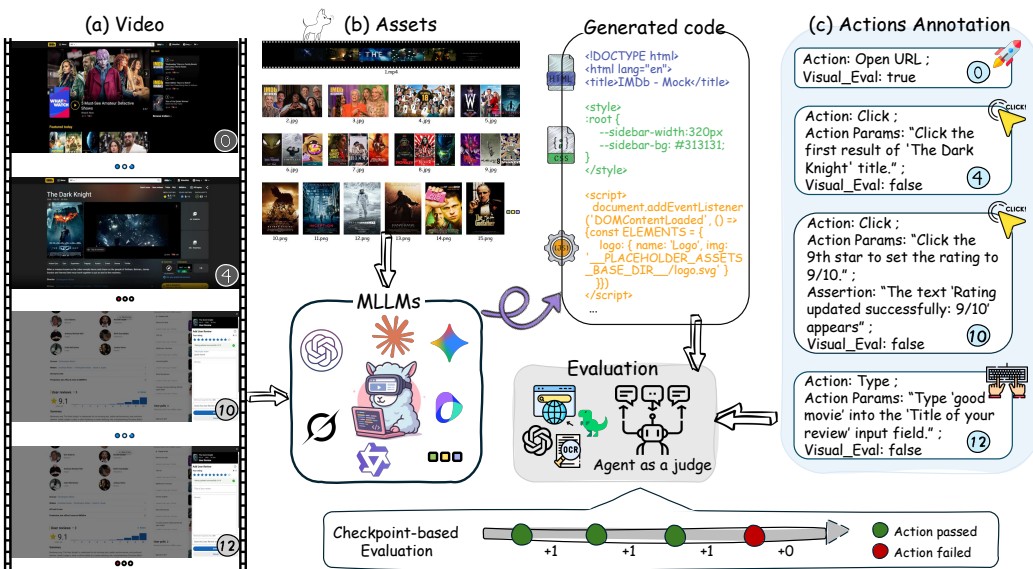

Figure 2: Overview of the IWR-Bench task and evaluation. The inputs to the model are (a) a user interaction video and (b) composite images of all static assets sniffed from the webpage. The evaluation employs an agent-as-judge framework (Zhuge et al., 2024), where an automated agent assesses the rendered page's interactivity by executing (c) a ground-truth action sequence and its visual fidelity through screenshot comparison.

moderate success, their capacity for synthesizing event-driven logic remains severely underdeveloped.

Our key contributions are:

- **A Benchmark for Interactive Webpage Reconstruction.** We introduce IWR-Bench, the first benchmark to formalize and evaluate Interactive Webpage Reconstruction (IWR) from video. It comprises 113 curated tasks from real-world websites, taxonomized along axes of domain, visual complexity, and interaction logic.
- **A Functionality-Centered Automated Evaluation Protocol.** We develop a robust evaluation protocol that employs a programmatic agent to assess functional correctness by executing ground-truth action sequences. Performance is quantified by two holistic metrics: the Interactive Functionality Score (IFS) and the Visual Fidelity Score (VFS).
- **An Extensive Evaluation and Analysis.** We conduct a comprehensive evaluation of 28 leading LVLMs, establishing strong initial baselines. The results reveal a critical performance gap between visual replication and functional implementation. Further analysis identifies systematic weaknesses in temporal reasoning and logic synthesis, outlining concrete directions for future research.

## 2  RELATED WORK

**Webpage Understanding.** Webpage understanding evolved from structural analysis based on DOM parsing to a subsequent multimodal perspective that jointly represents a page's visual and textual content (Furuta et al., 2023; Burns et al., 2023; Liu et al., 2024a). Large Vision-Language Models (LVLMs) have advanced webpage understanding by enabling a unified approach where a single model demonstrates strong performance across diverse downstream tasks, indicative of deep comprehension, such as element grounding (Team, 2025) and screen-based question answering (Wang et al., 2024b; Xu et al., 2024). Among these capabilities, generating code from a visual webpage representation is a key task where existing models have demonstrated strong performance (Beltramelli, 2018; Yun et al., 2024; Gui et al., 2025). With the enhanced capabilities of LVLMs in handling multiple images or videos (Bai et al., 2025; OpenAI, 2025; Guo et al., 2025; Comanici et al., 2025),

Operation Description (High

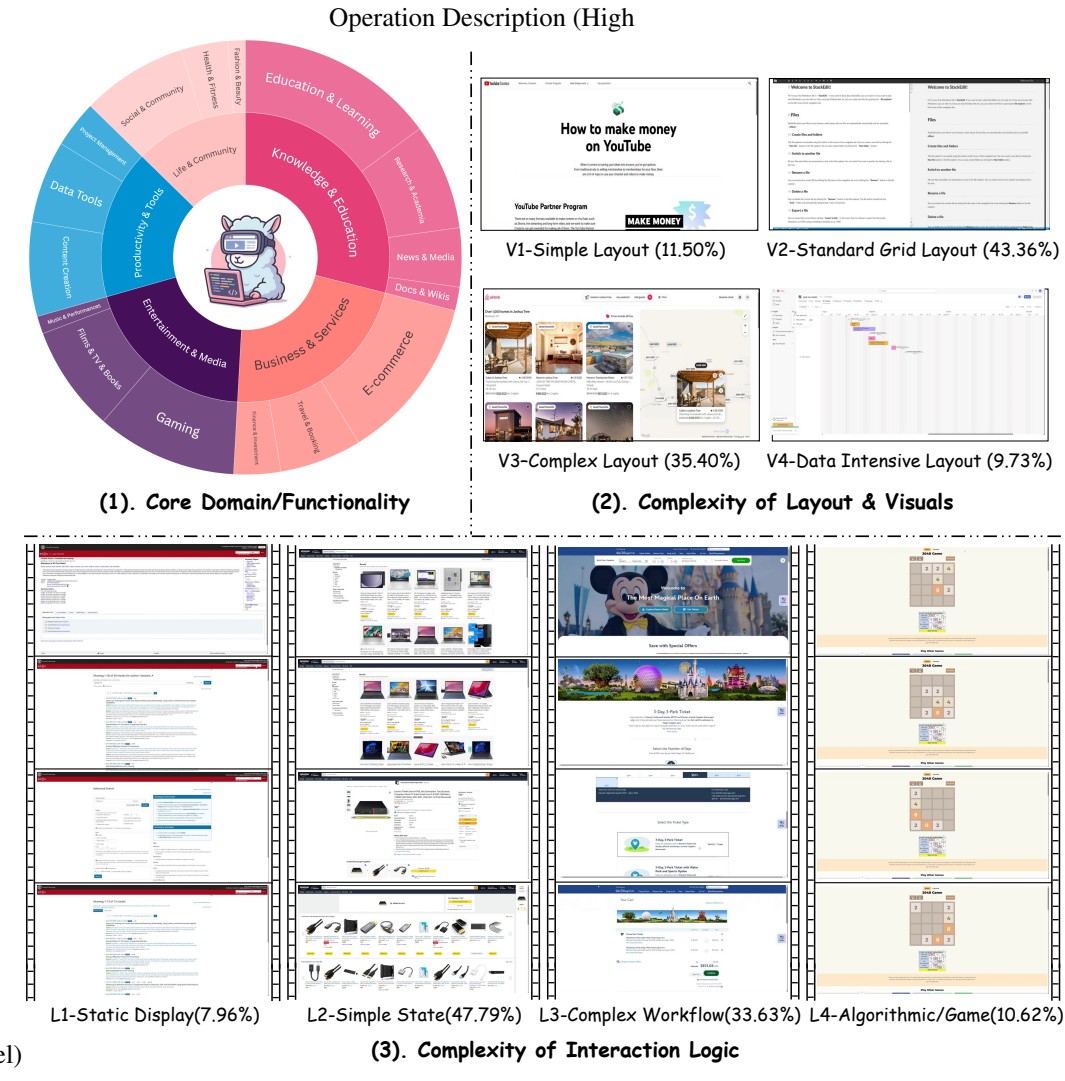

(1). **Core Domain/Functionality**

V1-Simple Layout (11.50%)    V2-Standard Grid Layout (43.36%)

V3-Complex Layout (35.40%)   V4-Data Intensive Layout (9.73%)

(2). **Complexity of Layout & Visuals**

L1-Static Display(7.96%)  L2-Simple State(47.79%)  L3-Complex Workflow(33.63%)  L4-Algorithmic/Game(10.62%)

Level)

(3). **Complexity of Interaction Logic**

Figure 3: An overview of the IWR-Bench taxonomy, which organizes tasks along three orthogonal axes: Domain, Visual Complexity, and Interaction Logic.

a logical extension of this capability is the generation of interactive webpages, moving beyond static layouts to better mimic real-world applications.

**LVLM Benchmarks.** The development of benchmarks for LVLMs has been driven by the rapid expansion of their capabilities, leading to evaluations of increasing complexity (Jimenez et al., 2024; Yang et al., 2024; Mialon et al., 2023; Lu et al., 2023). This progression is evident in the evolution from single-image comprehension to multi-image reasoning and video understanding (Yue et al., 2024; Li et al., 2023; Wang et al., 2024a; Liu et al., 2024b; Hu et al., 2025; Li et al., 2024; Fang et al., 2024; Fu et al., 2025; Ning et al., 2023; Chen et al., 2024; Yang et al., 2024; Lu et al., 2025). Concurrently, in the web domain, benchmarks have targeted either webpage understanding or static code generation from a single screenshot (Beltramelli, 2018; Laurençon et al., 2024; Yun et al., 2024; Si et al., 2024; Gui et al., 2025; Jiang et al., 2025; Awal et al., 2025; Xu et al., 2025), with works like IWR-Bench (Guo et al., 2024) creating more robust evaluation metrics for this task, while others like PairBench (Feizi et al., 2025) investigate the fundamental reliability of using models as evaluators. A recent advancement, Interaction2Code (Xiao et al., 2025), extends this by generating code from discrete interaction traces. However, such approaches primarily evaluate single-step, stateless events, rather than the complete, stateful workflows captured in continuous

video. Therefore, a critical disconnect exists between model capabilities for dynamic inputs and the benchmarks for interactive web generation.

## 3   IWR-BENCH

### 3.1   Task Definition and Structure

The Interactive Webpage Reconstruction (IWR) task challenges models to generate functional web code from observing user interactions. Formally, given a video $V = \{f_1, ..., f_n\}$ demonstrating user interactions and a set of static assets $A = \{a_1, ..., a_m\}$ from the original webpage, the model must generate code $C$ that reproduces both the visual appearance and interactive behavior observed in $V$. Each task instance in IWR-Bench comprises four key components:

- **Video Recording:** A screen capture documenting complete user interactions, preserving temporal dynamics and state transitions that define the webpage's behavior.
- **Static Web Assets:** All relevant images, icons, and videos necessary for reconstruction. To prevent models from leveraging prior knowledge based on semantic filenames (e.g., logo.png), all asset filenames are anonymized (e.g., renamed to asset_001.png) (Agrawal et al., 2018; Gurari et al., 2018). This forces the model to rely on visual matching and reasoning.
- **Action Trajectory:** A structured sequence $T = \{(a_i, p_i, d_i, v_i, l_i)\}_{i=1}^{k}$ where each action contains type $a_i$, parameters $p_i$, a natural language description $d_i$, a visual evaluation flag $v_i$, and logical assertions $l_i$ for verification.
- **Checkpoint Screenshots:** Stable-state images $S = \{s_1, ..., s_k\}$ capturing the visual state after each action. This ensures evaluation occurs on fully rendered pages rather than on transitional states.

### 3.2   Benchmark Construction

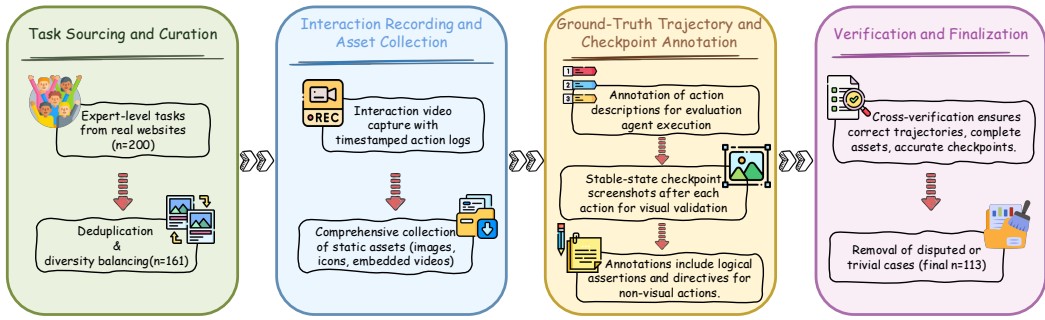

Figure 4: The overview of Benchmark construction.

Establishing and maintaining high standards for annotation quality and impartiality is a central design principle in the development of IWR-Bench. The process is shown in Figure 4.

**Task Sourcing and Curation.** The process begins with an initial set of 200 candidate tasks sourced from real-world websites by experts in web development. Each task is defined by a high-level goal and a URL to reflect common usage patterns. Through a rigorous curation process involving deduplication and balancing for diversity across predefined axes, such as domain and complexity (Figure 3), this set is reduced to a high-quality candidate pool of 161 tasks for annotation.

**Interaction Recording and Asset Collection.** For each curated task, interactions on the live website are performed by trained annotators and captured as screen recordings, while a browser extension concurrently records the action type $a_i$ and parameters $p_i$ for each action (Yun et al., 2024; Zhou et al., 2023). In parallel, all relevant static assets, such as images and icons, are collected via automated crawlers and manual inspection.

**Ground-Truth Trajectory and Checkpoint Annotation.** The raw recordings and action logs are converted into the final ground-truth representation. For each action, the logged type $a_i$ and param-

eters $p_i$ are augmented through a three-step annotation process: (1) a natural language description $d_i$ is authored to provide a clear instruction; (2) a visual evaluation flag $v_i$ is assigned, and a corresponding checkpoint screenshot $s_i$ is captured only when this flag is true, signifying a major visual state change; and (3) an optional logical assertion $l_i$ is defined where necessary to programmatically verify functional correctness, such as the appearance of a message or game logic.

**Verification.** Each annotated task undergoes a two-stage quality assurance process. First, a cross-verification review by a different annotator assesses trajectory correctness, asset completeness, and checkpoint fidelity, with a large model additionally used to verify the accuracy of logical assertions against the ground truth (Zheng et al., 2023; Gou et al., 2025). All identified discrepancies are rectified. Finally, disputed, ambiguous, or overly trivial tasks are filtered out, leaving a final collection of 113 verified tasks.

## 3.3 TAXONOMY AND STATISTICS

IWR-Bench organizes tasks along three orthogonal axes (Figure 3; see Appendix A for details): (1) Domain Coverage, which spans 5 major and 16 subcategories such as e-commerce and education to reflect real-world web diversity; (2) Visual Complexity, which scales from minimalist layouts to data-dense dashboards; and (3) Interaction Logic, which progresses from static content display to complex workflows and algorithmic game logic.

Table 1 presents key statistics. The benchmark includes 113 videos averaging 35.4 seconds, with 1,001 total actions across all sequences. Of these, 620 require visual evaluation and 403 include assertion checks, ensuring comprehensive assessment of both appearance and functionality. Tasks average 8.9 actions, with 10.62% targeting mobile interfaces, reflecting modern web usage patterns.

Table 1: Key Statistics of IWR-Bench.

| Statistic | Number |
| --- | --- |
| ***Video & Resolution Statistics*** | |
| Total Videos | 113 |
| - Short Videos ($\leq$20s) | 25 (22.1%) |
| - Medium Videos (20 $\sim$ 60s) | 72 (63.7%) |
| - Long Videos (>60s) | 16 (14.2%) |
| Video Duration (avg/max) | 35.4s / 172.9s |
| Unique Resolutions | 19 |
| - Mobile | 10.62% |
| ***Evaluation Statistics*** | |
| Total Actions in Sequences | 1001 |
| - Visual Evaluation | 620 |
| - Assertion Checks | 403 |
| Actions per Video (avg) | 8.9 |

## 3.4 COMPARISON TO OTHER BENCHMARKS

As detailed in Table 2, IWR-Bench addresses a critical gap between webpage reconstruction and video understanding benchmarks. Existing webpage reconstruction benchmarks either focus on static image-to-code tasks (e.g., Pix2Code, WebSight) or model interaction as stateless, single-step events without providing the necessary static assets for reconstruction (e.g., Interaction2Code). Conversely, general video understanding benchmarks (e.g., MVBench) are designed for comprehension tasks like Video QA, not code generation.

IWR-Bench overcomes these limitations by using videos of stateful, full-trajectory workflows from live websites. Interaction2CodeXiao et al. (2025), the most closely related benchmark, focuses on stateless, single-step events captured by image pairs from archived pages, whereas IWR-Bench requires temporal reasoning over complete user workflows with state transitions. Notably, while all existing webpage reconstruction benchmarks, including Interaction2Code, remove static assets to simplify the task, IWR-Bench provides all original assets (images, videos, icons) to enable realistic visual-to-asset grounding. This asset provision further allows the evaluation protocol to compute page-level visual similarity metrics, ensuring comprehensive assessment of reconstruction fidelity.

## 4 EVALUATION AND METRICS

### 4.1 EVALUATION PROTOCOL

The evaluation of generated code $C$ is conducted using a deterministic executor built upon the *browser-use* library (browser-use, 2025). This executor programmatically interacts with the rendered webpage by sequentially executing each pre-defined action $a_i$ from the ground-truth action

Table 2: Comparison of IWR-Bench with existing benchmarks. IWR-Bench is unique in its sourcing from live websites, video-based tasks, comprehensive interactive evaluation, and provision of static assets to create a realistic reconstruction task.

| Benchmark | Task Type | Data Source | Videos | Images (Checkpoints) | Asset Input | Desktop & Mobile | Interactive Evaluation |
|---|---|---|---|---|---|---|---|
| *(a) Webpage Reconstruction Benchmarks* | | | | | | | |
| Pix2Code (Beltramelli, 2018) | Image-to-Code | Synthesized | – | 1.7K | ✗ | ✓ | ✗ |
| DWCG (Yun et al., 2024) | Image-to-Code | Synthesized | – | 60K | ✗ | ✗ | ✗ |
| WebSight (Laurençon et al., 2024) | Image-to-Code | Synthesized | – | 2M | ✗ | ✗ | ✗ |
| Design2Code (Si et al., 2024) | Image-to-Code | C4 | – | 484 | ✗ | ✗ | ✗ |
| CC-HARD (Gui et al., 2025) | Image-to-Code | C4 | – | 128 | ✗ | ✗ | ✗ |
| ScreenCoder (Jiang et al., 2025) | Image-to-Code | Live Websites | – | 3K | ✗ | ✗ | ✗ |
| Interaction2Code (Xiao et al., 2025) | Images-to-Code | C4 & GitHub | – | 374 | ✗ | ✗ | ✓(Single-step) |
| *(b) Video Understanding Benchmarks* | | | | | | | |
| MMBench-Video (Fang et al., 2024) | Video QA | - | 609 | – | – | – | – |
| MVBench (Li et al., 2024) | Video QA | - | 20K | – | – | – | – |
| Video-MME (Fu et al., 2025) | Video QA | - | 900 | – | – | – | – |
| Video-MMMU (Hu et al., 2025) | Video QA | - | 300 | – | – | – | – |
| **IWR-Bench (Ours)** | **Video-to-Code** | **Live Websites** | **113** | **620*** | ✓ | ✓ | ✓(Full Trajectory) |

\* These are images used for evaluating visual fidelity across interaction states.

trajectory $T$. This design isolates the evaluation to code execution and removes any dependency on high-level task planning, thereby ensuring a stable and reproducible protocol.

The evaluation of the trajectory proceeds step-by-step. At each step $i$, the action $a_i$ is attempted. The action is considered a failure under two conditions: (1) it is operationally infeasible (e.g., a target element is not found), or (2) its corresponding logical assertions $l_i$ are not satisfied.

For logical assertion verification, an MLLM judge, specifically **Gemini-2.5-Pro** (Comanici et al., 2025), is employed to analyze screenshots of the page state before and after an action to determine its correctness. The prompt for this judge is detailed in Appendix E. Upon the successful completion of an action, a new screenshot is captured. If the visual evaluation flag $v_i$ for this step is true, the new screenshot undergoes a visual fidelity assessment. This assessment is based on a composite score that integrates OCR-based text similarity (Cui et al., 2025), DINO-based structural similarity (Oquab et al., 2023), and a high-level evaluation also conducted by **Gemini-2.5-Pro** (see Appendix E for the prompt). Actions where $v_i$ is false, which typically involve insignificant or stochastic visual changes, are omitted from this visual assessment phase.

## 4.2 METRICS

Model performance is quantified through a hierarchy of metrics designed to measure functional correctness, visual fidelity, and overall task completion.

**Interactive Functionality Score (IFS).** This metric measures a model's ability to generate functionally correct code. An action $a_i$ from the trajectory $T$ is considered successful if and only if it executes without operational errors and all associated logical assertions $l_i$ are satisfied, as determined by the protocol in Section 4.1. The IFS is defined as the ratio of successfully completed actions ($N_{\text{succ}}$) to the total number of actions ($N_{\text{total}}$).

$$\text{IFS} = \frac{N_{\text{succ}}}{N_{\text{total}}} \tag{1}$$

**Visual Fidelity Score (VFS).** The VFS assesses the visual quality of the rendered user interface. This score is computed exclusively over checkpoints that were successfully reached and have the visual evaluation flag enabled ($v_i = \text{true}$). Let $I_{v,\text{succ}}$ be the set of indices for these qualifying checkpoints. The score for each checkpoint $i \in I_{v,\text{succ}}$ is a weighted combination of two components: a *Low-level Visual Score* ($S_{\text{LVS},i}$), which averages an OCR-based Levenshtein similarity and a DINO-based cosine similarity, and a *High-level Visual Score* ($S_{\text{HVS},i}$), which is a holistic assessment from the MLLM judge. The final VFS is the macro-average of these checkpoint scores. The weight $w$ is set to 0.5 based on validation studies (Section 5.4).

$$\text{VFS} = \frac{1}{|I_{v,\text{succ}}|} \sum_{i \in I_{v,\text{succ}}} (w \cdot S_{\text{LVS},i} + (1 - w) \cdot S_{\text{HVS},i}) \tag{2}$$

**Final Score.** The Final Score is defined by combining the IFS and VFS with fixed weights. For steps where actions cannot be executed, no images are available to compute visual similarity scores. An alternative weighting scheme based on the ratio of successful ($N_{\text{succ}}$) to total ($N_{\text{total}}$) steps was explored, but it proved ineffective for differentiating model performance. Therefore, a simple weighted combination is adopted with the weighting factor $\alpha$ set to 0.7 (see Section 5.4). By assigning substantial weight to the IFS component, the impact of unreachable states on overall evaluation is appropriately reflected. All reported scores are macro-averaged across the entire benchmark.

$$\text{Final Score} = \alpha \cdot \text{IFS} + (1 - \alpha) \cdot \text{VFS} \tag{3}$$

## 5 EXPERIMENTS

### 5.1 EVALUATION SETUP

**Evaluation Models.** The evaluation is conducted on a diverse set of 28 leading Large Vision-Language Models (LVLMs) to establish a comprehensive performance baseline on IWR-Bench. This selection encompasses both proprietary and open-source models, as well as specialized video understanding models. The full list of evaluated models and their performance is detailed in Table 3.

**Implementation Details.** For each task in IWR-Bench, models are provided with the user interaction video and a composite image of all crawled static assets. To accommodate models without native video support, each video is sampled at 1 fps, with the number of frames capped at 64. Videos exceeding 64 seconds are uniformly downsampled to meet this limit. The video (or its sampled frames) and the composite image are arranged as a sequential, multi-image input. The task is to generate a single, self-contained HTML file that integrates all necessary CSS and JavaScript to replicate the observed webpage. All other inference parameters utilize the default settings recommended by the model providers. The complete prompt templates are detailed in Appendix E.

### 5.2 MAIN RESULTS

The comprehensive evaluation results on IWR-Bench are presented in Table 3. The findings reveal a clear performance landscape, highlighting the substantial difficulty of the task and surfacing several key observations regarding current model capabilities, with case studies provided in Appendix F.

**A Clear Performance Hierarchy Is Observed Across Model Categories.** The results on IWR-Bench show a pronounced performance stratification across model categories. Proprietary multimodal large language models are positioned in the upper echelon, with GPT-5 obtaining the highest Final Score (36.35). This is followed by a competitive cluster that includes Claude-Sonnet-4 (thinking) (34.62), Claude-Opus-4 (thinking) (34.13), Doubao-seed-1.6 (34.02), and Claude-Sonnet-4 (34.00). The top-performing open-source model, Qwen3-VL (thinking), has a score of 31.15. This score is lower than that of the leading proprietary group but surpasses several mid-tier proprietary entries, such as GPT-4o (latest) (29.55). At the lower end of the performance spectrum, video-specialized models like VideoLLaMA3-7B (13.67) and InternVideo-2.5-Chat-8B (10.07) are found. This hierarchy indicates that general multimodal reasoning and code generation capabilities are more critical for success on IWR-Bench than specialized video-processing architectures.

**Interactive Functionality Remains the Primary Performance Bottleneck.** A substantial performance gap exists between static visual replication and dynamic functionality implementation. This gap is reflected in the consistently higher Visual Fidelity Scores (VFS) compared to the Interactive Functionality Scores (IFS). For instance, GPT-5 obtains the highest visual metrics (LVS 68.29, HVS 60.21, VFS 64.25), yet its corresponding IFS is only 24.39. A similar pattern is observed for Claude-Sonnet-4, which has the second-highest VFS (61.34) but an IFS of only 22.29. The difficulty of this task is further underscored by the low absolute IFS values, with the highest score remaining below 25, highlighting that interactive webpage reconstruction is a largely unsolved problem.

**Reasoning Enhancement Provides Consistent but Moderate Gains.** Consistent but moderate performance improvements are observed when using reasoning-enhanced inference. For instance, the "thinking" variant of Claude-Sonnet-4 shows higher performance in both Final Score (34.62 vs. 34.00) and IFS (23.65 vs. 22.29). A similar trend is noted for Claude-Opus-4 (Final 34.13 vs. 33.33; IFS 23.61 vs. 21.83) and Gemini-2.5-Pro (Final 30.36 vs. 30.31; IFS 21.65 vs. 20.51).

Table 3: Main evaluation results on IWR-Bench. Models are grouped by category and sorted by Final Score. Reasoning-enhanced ('thinking') model variants are highlighted in gray. The best result in each column is **bolded**, and the second-best is underlined.

| Model | Low-level Visual Score | High-level Visual Score | Visual Fidelity Score | Interactive Functionality Score | Final Score |
|---|---|---|---|---|---|
| *Proprietary MLLMs* | | | | | |
| GPT-5 (OpenAI, 2025) | **68.29** | **60.21** | **64.25** | **24.39** | **36.35** |
| Claude-Sonnet-4 (thinking) (anthropic, 2025b) | 64.90 | 55.51 | 60.20 | 23.65 | 34.62 |
| Claude-Opus-4 (thinking) (anthropic, 2025a) | 63.53 | 53.80 | 58.67 | 23.61 | 34.13 |
| Doubao-seed-1.6 (bytedance, 2025) | 65.95 | 55.62 | 60.79 | 22.55 | 34.02 |
| Claude-Sonnet-4 (anthropic, 2025b) | 65.75 | 56.92 | 61.34 | 22.29 | 34.00 |
| Claude-Opus-4 (anthropic, 2025a) | 65.23 | 55.13 | 60.18 | 21.83 | 33.33 |
| GPT-5-mini (OpenAI, 2025) | 63.36 | 50.25 | 56.81 | 23.18 | 33.27 |
| GPT-4.1 (OpenAI, 2025) | 63.07 | 54.63 | 58.85 | 20.48 | 31.99 |
| Gemini-2.5-Pro (thinking) (Comanici et al., 2025) | 54.52 | 46.83 | 50.67 | 21.65 | 30.36 |
| Gemini-2.5-Pro (Comanici et al., 2025) | 57.46 | 48.91 | 53.18 | 20.51 | 30.31 |
| GPT-4o (latest) (Hurst et al., 2024) | 63.39 | 51.71 | 57.55 | 17.55 | 29.55 |
| Gemini-2.5-Flash (Comanici et al., 2025) | 47.53 | 37.75 | 42.64 | 19.88 | 26.71 |
| GPT-5-nano (OpenAI, 2025) | 53.49 | 35.70 | 44.59 | 18.17 | 26.10 |
| Grok-4 (X.ai, 2025) | 48.95 | 30.54 | 39.74 | 19.44 | 25.53 |
| GPT-4o (0806) (Hurst et al., 2024) | 54.03 | 39.83 | 46.93 | 15.87 | 25.19 |
| Doubao-seed-1.6-flash (bytedance, 2025) | 45.49 | 32.06 | 38.78 | 16.34 | 23.07 |
| Gemini-2.5-Flash-Lite (Comanici et al., 2025) | 28.95 | 19.05 | 24.00 | 13.29 | 16.50 |
| *Open-Source MLLMs* | | | | | |
| Qwen3-VL (thinking) (QwenTeam, 2025) | 58.55 | 46.13 | 52.34 | 22.07 | 31.15 |
| Qwen2.5-VL-72B (Bai et al., 2025) | 47.83 | 28.25 | 38.04 | 17.42 | 23.61 |
| Qwen2.5-VL-32B (Bai et al., 2025) | 39.36 | 23.30 | 31.33 | 16.50 | 20.95 |
| Keye-VL-1.5-8B (Yang et al., 2025) | 30.81 | 15.49 | 23.15 | 16.06 | 18.18 |
| MiniCPM-V-4.5 (Yu et al., 2025) | 31.18 | 15.41 | 23.29 | 15.58 | 17.89 |
| Qwen2.5-VL-7B (Bai et al., 2025) | 28.92 | 12.20 | 20.56 | 13.28 | 15.47 |
| Kimi-VL (thinking) (Team et al., 2025b) | 26.18 | 12.23 | 19.20 | 12.04 | 14.19 |
| Mimo-VL-7B (Team et al., 2025a) | 23.28 | 4.99 | 14.14 | 10.57 | 11.64 |
| GLM-4.5V (Team et al., 2025c) | 16.31 | 10.52 | 13.41 | 10.11 | 11.10 |
| *Open-Source Video-Specialized LMs* | | | | | |
| VideoLLaMA3-7B (Zhang et al., 2025) | 31.29 | 11.86 | 21.58 | 10.29 | 13.67 |
| InternVideo-2.5-Chat-8B (Wang et al., 2025) | 17.27 | 3.33 | 10.30 | 9.97 | 10.07 |

This evidence indicates that while enhanced reasoning acts as a useful refinement, the base model's capability remains the primary factor determining the performance ceiling on IWR-Bench.

## 5.3 PERFORMANCE ANALYSIS ACROSS TASK DIMENSIONS

A fine-grained analysis (detailed in Appendix B) reveals distinct performance patterns. The synthesis of event-driven functionality is the primary bottleneck, evidenced by a sharp performance drop from static (L1) to interactive (L2-L4) tasks (Table 4). Models also struggle with highly structured layouts (Table 5). Performance varies by domain, with relative strength in "Entertainment & Media" (Table 6), pointing to structured code generation and state management as key research directions.

## 5.4 VALIDATION OF THE EVALUATION PROTOCOL

The robustness and reliability of the evaluation protocol are validated through a rigorous, two-part analysis that addresses both the metric parameters and the agent-as-a-judge methodology (Zheng et al., 2023; Gou et al., 2025; Maaz et al., 2023). First, the weighting coefficients ($w$ and $\alpha$) for the scoring metrics are determined through a human alignment study, with the detailed procedure and results presented in Appendix C. Second, the agent-as-a-judge framework is validated through a multi-stage process. This process includes a meticulous cross-verification of annotated action trajectories (Section 3.2), automated verification of logical assertions using an MLLM-based judge(Comanici et al., 2025), and manual inspection of the agent's operational fidelity. For the manual inspection, three PhD students observed the agent's execution on 100 randomly sampled, model-generated web-pages, with the browser's headless mode disabled to compare on-screen behavior against evaluation

logs. Failures in the agent's evaluation were observed in only three instances. These issues typically stemmed from ambiguous element descriptors (e.g., buttons with identical names), which required a more precise locator (`d_i`). All identified discrepancies were subsequently rectified.

## 6 CONCLUSION

This paper introduces IWR-Bench, the first benchmark designed to evaluate Interactive Webpage Reconstruction from video. Through an automated agent-as-a-judge evaluation protocol, performance is quantified using two metrics: the IFS and the VFS. Comprehensive evaluations on 28 LVLMs reveal a stark disparity between visual replication and functional implementation. While models achieve moderate success in reconstructing static appearance (VFS), their ability to generate correct, event-driven logic remains critically limited, as shown by low IFS scores across the board. This finding indicates that the primary bottleneck for current models is not visual understanding but the synthesis of complex interaction logic. IWR-Bench thus establishes a challenging new frontier for vision-language research, highlighting the need for future work to focus on temporal reasoning, dynamic asset binding, and robust code synthesis to create truly functional web applications.

# 7 ACKNOWLEDGMENT

The research was supported by Shanghai Artificial Intelligence Laboratory.

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

## A  MULTI-DIMENSIONAL TASK TAXONOMY

To move beyond a monolithic view of difficulty and enable a fine-grained analysis of model capabilities, we developed a three-dimensional taxonomy to classify each task. This taxonomy categorizes tasks along the orthogonal axes of Interaction Complexity, Visual Complexity, and Application Domain, providing a structured framework to understand the specific challenges inherent in each task and to diagnose model failure modes with high precision.

**Interaction Complexity (L1-L4)**  This first axis categorizes tasks based on the depth of logical and temporal understanding required for successful reconstruction.

- **L1: Static Content Consumption.** Tasks involve passive information consumption, primarily requiring correct handling of vertical scrolling to reconstruct long pages that extend beyond a single viewport (e.g., browsing a blog post or a project's README).
- **L2: Simple State Manipulation.** Tasks feature components that manage local state, such as filtering e-commerce results, switching between on-page tabs, or expanding/collapsing accordion menus. This level tests the generation of basic client-side event handlers.
- **L3: Complex Workflow Interaction.** These tasks involve multi-step, sequential interactions where state is passed between components, such as a multi-step product configurator or an online booking process. This tests understanding of application logic and inter-component communication.
- **L4: Algorithmic/Game Logic.** The most complex level requires the model to reverse-engineer and implement a set of rules or algorithms, such as an online calculator, a text-based puzzle, or a simple game like 2048.

**Visual Complexity (V1-V4)**  This second axis captures the static challenge of rendering the webpage's appearance, focusing on its layout and styling.

- **V1: Minimalist Layouts.** Simple, single- or two-column structures with standard element alignment, typical of documentation or text-heavy sites.
- **V2: Standard Grid-based Layouts.** Organized grid systems are used in e-commerce or news portals, featuring numerous but regularly arranged elements.
- **V3: Asymmetric & Modern Layouts.** Visually-driven designs with complex CSS, such as overlapping elements, parallax scrolling, and non-standard component shapes.
- **V4: Data-Dense Layouts.** Dashboards or admin panels with a high density of information presented in charts, tables, and data cards, testing the ability to generate precise, repetitive structures.

**Application Domain**  To ensure our benchmark reflects the breadth of real-world web applications, the third axis classifies tasks by their Application Domain. Drawing inspiration from established taxonomies in web-centric agent research (Gou et al., 2025), we group tasks into five high-level domains: **Commerce & Services** (e-commerce, booking, finance), **Knowledge & Education** (academic sites, news portals, documentation), **Productivity & Tools** (calculators, project management boards), **Entertainment & Media** (games, streaming platforms), and **Lifestyle & Community** (social forums, blogs). This classification guarantees that models are evaluated across a diverse spectrum of functionalities and visual paradigms. For instance, a task involving filtering products on an e-commerce site is tagged as [L2, V2, Commerce], while a task requiring the reconstruction of a simple browser game is tagged as [L4, V1, Entertainment]. This multi-dimensional labeling allows us to analyze whether a model's performance correlates with certain types of interactions, visual styles, or application contexts.

## B  DETAILED EXPERIMENTAL RESULTS

This appendix provides a comprehensive breakdown of model performance on IWR-Bench across the three classification axes defined in the taxonomy: Application Domain (Table 6), Interaction Logic Complexity (Table 4), and Visual Complexity (Table 5). For each task category, the Final Score is reported.

Table 4: Final Score breakdown by Interaction Logic Complexity.

| Model | Static Content Consumption (L1) | Simple State Manipulation (L2) | Complex Workflow Interaction (L3) | Algorithmic/Game Logic (L4) |
|---|---|---|---|---|
| *Proprietary MLLMs* | | | | |
| GPT-5 | 61.85 | 35.43 | 35.12 | 25.26 |
| Claude-Sonnet-4 (thinking) | 65.75 | 33.88 | 31.05 | 25.86 |
| Claude-Opus-4 (thinking) | 61.78 | 31.52 | 32.57 | 30.07 |
| Doubao-seed-1.6 | 63.69 | 34.03 | 30.12 | 24.08 |
| Claude-Sonnet-4 | 68.36 | 31.04 | 32.86 | 25.15 |
| Claude-Opus-4 | 66.88 | 30.85 | 31.30 | 25.76 |
| GPT-5-mini | 56.83 | 32.10 | 31.38 | 26.84 |
| GPT-4.1 | 51.45 | 30.32 | 31.98 | 24.78 |
| Gemini-2.5-Pro (thinking) | 68.96 | 26.88 | 28.15 | 23.83 |
| Gemini-2.5-Pro | 59.58 | 31.51 | 23.95 | 22.58 |
| GPT-4o (latest) | 48.61 | 29.09 | 27.53 | 23.75 |
| Gemini-2.5-Flash | 56.57 | 24.95 | 25.53 | 15.00 |
| GPT-5-nano | 46.23 | 25.57 | 24.31 | 19.02 |
| Grok-4 | 48.37 | 24.94 | 22.76 | 19.53 |
| GPT-4o (0806) | 38.59 | 24.95 | 24.65 | 17.90 |
| Doubao-seed-1.6-flash | 42.77 | 22.14 | 22.65 | 13.88 |
| Gemini-2.5-Flash-Lite | 35.34 | 15.61 | 13.87 | 14.70 |
| *Open-Source MLLMs* | | | | |
| Qwen3-VL (thinking) | 51.05 | 30.43 | 29.86 | 23.60 |
| Qwen2.5-VL-72B | 45.35 | 22.83 | 21.90 | 15.46 |
| Qwen2.5-VL-32B | 37.64 | 20.45 | 18.80 | 17.48 |
| Keye-VL-1.5-8B | 46.85 | 19.47 | 12.11 | 10.13 |
| MiniCPM-V-4.5 | 37.30 | 18.02 | 15.27 | 11.10 |
| Qwen2.5-VL-7B | 28.58 | 15.99 | 13.91 | 8.21 |
| Kimi-VL (thinking) | 23.61 | 15.45 | 10.63 | 12.75 |
| Mimo-VL-7B | 22.23 | 12.32 | 9.36 | 7.87 |
| GLM-4.5V | 18.56 | 11.02 | 9.77 | 10.08 |
| *Open-Source Video-Specialized LMs* | | | | |
| VideoLLaMA3-7B | 23.11 | 14.15 | 12.31 | 8.77 |
| InternVideo-2.5-Chat-8B | 27.89 | 9.81 | 7.68 | 5.42 |

## C  METRIC PARAMETER VALIDATION

The VFS and Final Score metrics rely on the weighting coefficients $w$ and $\alpha$. These parameters are determined and validated through a human alignment study. A sample of 60 evaluation instances is constructed by selecting outputs from three randomly chosen models for each of 20 randomly selected tasks from IWR-Bench. Each instance is assessed by five PhD-level students on two dimensions: visual fidelity and overall quality. To determine the optimal parameters, a grid search is performed over the discrete set $\{0.1, 0.2, \ldots, 0.9\}$ for both $w$ and $\alpha$. The value that maximizes the Spearman's $\rho$ correlation between the automated scores and the aggregated human judgments is selected. For the VFS metric, the peak correlation ($\rho = 0.57$) is observed at $w = 0.5$, indicating an equal weighting between LVS and HVS. For the Final Score, the maximum correlation with human overall judgment ($\rho = 0.65$) is achieved with $\alpha = 0.7$, empirically validating the decision to weigh functionality (IFS) more heavily than visual fidelity (VFS).

## D  TASK AND ACTION REPRESENTATION

Each task in IWR-Bench is formally defined by an 'action_sequence', a structured list of discrete actions that an automated agent must perform to validate the reconstructed webpage. This representation standardizes the evaluation process. We defined a vocabulary of atomic actions, including **Click(description)**, **Type(key, description)**, **Scroll(direction, amount, description)**, and **Press(key, description)**. A crucial design choice is the use of a natural language 'description' field for targeting elements instead of unstable positional coordinates (e.g., "Click the primary 'Submit' button" instead of "Click at (x:120, y:350)"). This makes the evaluation robust to minor layout variations in the generated code and tests a more semantic understanding of the page structure, both for the model during generation and the agent during evaluation.

Table 5: Final Score breakdown by Visual Complexity.

| Model | Minimalist Layouts (V1) | Standard Grid-based Layouts (V2) | Asymmetric & Modern Layouts (V3) | Data-Dense Layouts (V4) |
|---|---|---|---|---|
| *Proprietary MLLMs* | | | | |
| GPT-5 | 44.77 | 30.73 | 43.77 | 26.05 |
| Claude-Sonnet-4 (thinking) | 40.05 | 31.12 | 40.08 | 25.26 |
| Claude-Opus-4 (thinking) | 37.01 | 32.26 | 38.95 | 22.97 |
| Doubao-seed-1.6 | 41.55 | 30.56 | 38.03 | 26.97 |
| Claude-Sonnet-4 | 37.33 | 30.28 | 39.95 | 26.25 |
| Claude-Opus-4 | 35.77 | 30.79 | 38.34 | 24.77 |
| GPT-5-mini | 38.76 | 29.29 | 38.44 | 26.75 |
| GPT-4.1 | 35.24 | 28.96 | 36.89 | 25.32 |
| Gemini-2.5-Pro (thinking) | 31.59 | 25.77 | 36.99 | 25.82 |
| Gemini-2.5-Pro | 37.20 | 25.17 | 36.51 | 23.27 |
| GPT-4o (latest) | 31.89 | 26.14 | 34.64 | 24.41 |
| Gemini-2.5-Flash | 32.21 | 21.80 | 34.13 | 16.28 |
| GPT-5-nano | 29.21 | 24.39 | 28.96 | 20.37 |
| Grok-4 | 33.73 | 22.11 | 30.01 | 17.18 |
| GPT-4o (0806) | 29.88 | 23.54 | 27.41 | 19.62 |
| Doubao-seed-1.6-flash | 30.23 | 22.76 | 22.87 | 17.27 |
| Gemini-2.5-Flash-Lite | 19.64 | 15.44 | 19.31 | 8.31 |
| *Open-Source MLLMs* | | | | |
| Qwen3-VL (thinking) | 38.14 | 28.74 | 33.37 | 26.25 |
| Qwen2.5-VL-72B | 28.68 | 21.14 | 28.31 | 12.48 |
| Qwen2.5-VL-32B | 27.56 | 17.90 | 24.06 | 16.15 |
| Keye-VL-1.5-8B | 24.75 | 15.65 | 20.85 | 12.74 |
| MiniCPM-V-4.5 | 28.24 | 16.11 | 18.87 | 10.79 |
| Qwen2.5-VL-7B | 20.48 | 12.68 | 18.52 | 11.48 |
| Kimi-VL (thinking) | 18.57 | 14.22 | 14.21 | 9.26 |
| Mimo-VL-7B | 14.68 | 11.42 | 12.42 | 6.71 |
| GLM-4.5V | 12.39 | 10.66 | 13.34 | 4.20 |
| *Open-Source Video-Specialized LMs* | | | | |
| VideoLLaMA3-7B | 18.81 | 13.55 | 13.69 | 8.56 |
| InternVideo-2.5-Chat-8B | 13.36 | 11.07 | 9.09 | 5.57 |

# E PROMPTS

A standardized system prompt is employed for all models and tasks in IWR-Bench to ensure a fair evaluation. This prompt defines clear requirements for the task, output format, and operational constraints. Such a design minimizes ambiguity and helps isolate the core code generation capabilities of each model. The complete prompt template is detailed in Figure 5.

The evaluation relies on a large multimodal model guided by two distinct prompts. To assess the similarity between generated and reference webpages, a prompt template is utilized (Figure 6). This template instructs the model to perform both quantitative and qualitative evaluations. For logical assertion verification, a separate prompt, presented in Figure 7, is employed to determine the correctness of an action.

# F CASE STUDY

This section presents a selection of representative tasks from the IWR-Bench to illustrate the diversity of challenges encompassed by our benchmark. The input of each case includes a webpage operation video and the static resources involved in the webpage. Then the web pages generated by different multimodal large models and the corresponding interaction results are displayed. Then we provide a detailed analysis of these representative cases, corresponding to the figures presented below. Each analysis breaks down the task's objectives, its position within our taxonomy, and the specific model behaviors observed, illustrating how our benchmark facilitates a fine-grained diagnosis of model capabilities.

Table 6: Final Score breakdown by Application Domain.

| Model | Business & Services | Entertainment & Media | Knowledge & Education | Life & Community | Productivity & Tools |
|---|---|---|---|---|---|
| *Proprietary MLLMs* | | | | | |
| GPT-5 | 39.37 | 47.24 | 37.05 | 22.80 | 28.53 |
| Claude-Sonnet-4 (thinking) | 39.74 | 41.02 | 32.60 | 24.86 | 31.15 |
| Claude-Opus-4 (thinking) | 35.54 | 42.48 | 34.91 | 24.21 | 28.56 |
| Doubao-seed-1.6 | 37.82 | 43.19 | 32.14 | 22.44 | 30.20 |
| Claude-Sonnet-4 | 38.84 | 42.09 | 32.22 | 25.57 | 27.58 |
| Claude-Opus-4 | 38.64 | 43.25 | 30.51 | 22.03 | 28.09 |
| GPT-5-mini | 37.16 | 45.11 | 30.29 | 21.38 | 28.36 |
| GPT-4.1 | 33.58 | 45.35 | 27.61 | 17.28 | 32.65 |
| Gemini-2.5-Pro (thinking) | 31.49 | 41.67 | 30.31 | 15.82 | 26.34 |
| Gemini-2.5-Pro | 33.66 | 36.93 | 30.74 | 19.03 | 25.55 |
| GPT-4o (latest) | 36.33 | 32.35 | 28.97 | 20.01 | 25.42 |
| Gemini-2.5-Flash | 29.16 | 40.60 | 21.38 | 15.27 | 25.83 |
| GPT-5-nano | 30.61 | 34.29 | 22.67 | 16.50 | 23.69 |
| Grok-4 | 27.71 | 32.06 | 26.47 | 15.00 | 19.85 |
| GPT-4o (0806) | 27.60 | 32.05 | 23.97 | 15.54 | 23.29 |
| Doubao-seed-1.6-flash | 25.37 | 31.07 | 20.17 | 14.21 | 22.36 |
| Gemini-2.5-Flash-Lite | 18.38 | 24.93 | 10.13 | 9.42 | 20.55 |
| *Open-Source MLLMs* | | | | | |
| Qwen3-VL (thinking) | 37.36 | 35.00 | 29.57 | 17.71 | 31.19 |
| Qwen2.5-VL-72B | 28.71 | 31.20 | 20.89 | 14.29 | 19.88 |
| Qwen2.5-VL-32B | 24.05 | 28.58 | 20.67 | 9.80 | 16.73 |
| Keye-VL-1.5-8B | 19.27 | 26.29 | 16.42 | 9.95 | 16.50 |
| MiniCPM-V-4.5 | 17.10 | 26.63 | 17.38 | 11.21 | 14.63 |
| Qwen2.5-VL-7B | 17.11 | 21.96 | 16.69 | 4.67 | 11.61 |
| Kimi-VL (thinking) | 16.77 | 17.58 | 13.87 | 9.59 | 10.83 |
| Mimo-VL-7B | 12.90 | 14.64 | 12.11 | 4.33 | 11.08 |
| GLM-4.5V | 10.78 | 18.20 | 10.10 | 2.88 | 11.10 |
| *Open-Source Video-Specialized LMs* | | | | | |
| VideoLLaMA3-7B | 14.36 | 16.44 | 14.44 | 9.21 | 11.51 |
| InternVideo-2.5-Chat-8B | 10.77 | 15.77 | 9.18 | 2.90 | 9.36 |

**Case 1 Analysis: E-commerce Workflow Simulation.** Our first case study, classified as [L2, V2, E-commerce], simulates a fundamental e-commerce user journey to test a model's ability to handle sequential state manipulations within a standard visual structure. The task requires the model to replicate a workflow involving filtering a product grid by a specific brand, sorting the filtered results by price and adding a selected item to the shopping cart.

As illustrated in Figure 8, this task effectively exposes different failure modes in different models. On the left, Claude-Sonnet-4 demonstrates good capabilities in static replication and simple state management. It accurately renders the initial layout and correctly implements the action for filtering and sorting. However, its failure occurs at the final "add to cart" step. The right side shows the result of GPT-5. This case likely involved too many static resources, causing the product list on the initial page to fail to render successfully. By pinpointing these different stages of failure, the benchmark provides a granular diagnosis of each model's specific strengths and weaknesses in front-end code generation.

**Case 2 Analysis: Algorithmic Logic Reconstruction.** This case study moves to the highest level of our interaction complexity scale, L4, to assess a model's capacity for algorithmic reasoning. Classified as [L4, V2, Gaming], the task requires the model to reverse-engineer and implement the complete set of rules for a simple browser game (e.g., 2048) based solely on observing its behavior in the input video. The visual complexity is simple (V2), deliberately shifting the evaluation focus from layout replication to the correctness of the underlying algorithmic logic. The The core challenge is to deduce and codify the game's state-transition functions, including tile movement, merging logic, and the spawning of new tiles.

---

**Prompt template for the IWR-Bench**

You are an expert front-end developer. Your task is to create a pixel-perfect replica of a website from a video.

Generate a single 'index.html' file that contains all HTML, CSS, and JavaScript necessary to replicate the UI, content, and interaction features shown. The webpage resolution in the video is <resolution>.

Instructions:

1. Single File Output: All HTML, CSS, and JS must be in one 'index.html' file.

2. If backend logic is implied, mock it in JS with static data (e.g., a JS array for a fake API call).

3. Assets(Images and Videos in the webpage):

   − All images must use the provided stitched image assets.

   − The 'src' attribute must start with the literal, unchanging string '__PLACEHOLDER_ASSETS_BASE_DIR__/', followed by the actual filename identified from the stitched image.

     − For example: 'src="__PLACEHOLDER_ASSETS_BASE_DIR__/asset001.svg"'.

     − '' tags must include 'width' and 'height' attributes.

     − The provided stitched image assets are before the video.

4. No External Dependencies: The generated code must be entirely self-contained. No External Libraries and no External Fonts.

5. Final Response: Return **only the complete HTML code** in a single "'html code block, with no additional text or explanations.

Figure 5: Prompt template for the IWR-Bench

---

**Prompt for evaluating HVS**

You are an expert Webpage Evaluator. Your task is to provide a quantitative and qualitative assessment of the similarity between a generated webpage and a reference webpage.

The default score is 0.

Evaluation Format:

—

Comments:

-Layout (10 points): ${comment and subscore}

-Elements (15 points): ${comment and subscore}

-Content and Text (40 points): ${comment and subscore}

-Style (15 points): ${comment and subscore}

-Overall (20 points): ${comment and subscore}

Score: ${final score}/100

Figure 6: Prompt for evaluating HVS

---

As illustrated in Figure 9, the left side is the result of Grok-4, which can successfully reproduce the 2048 game logic from the input video. However, Qwen2.5-VL-72B failed to merge the corresponding blocks after inputting '↑'. This type of task requires a high level of logical reasoning ability from the model and is a significant challenge.

**Case 3 Analysis: Long-Context Fidelity and Fine-Grained Visual Detail.**   This case study, classified as [L1, V3, E-commerce], is designed to stress-test a model's visual fidelity on multiple fronts. While the interaction is simple (L1, passive scrolling), the task's difficulty lies in three distinct challenges: (1) maintaining structural integrity across a long page, (2) correctly handling diverse media assets, including an embedded video, and (3) achieving fine-grained visual accuracy, particularly with small, repetitive elements like icons. This multi-faceted task evaluates not just broad layout re-

> **Prompt used to determine whether the assertion is correct**
>
> Please compare two webpage screenshots (Image 1 is the previous step, Image 2 is the current step) and determine whether the following assertion is true:
> Assertion: {assertion}
> Return JSON format without any additional information:
> {{
> "think": "the thinking process",
> "result": "Yes—No"
> }}

Figure 7: Prompt used to determine whether the assertion is correct

construction but also the model's attention to detail and its ability to precisely match visual elements to the provided stitched assets.

As illustrated in Figure 10, Gemini-2.5-pro and GPT-5 can both restore relatively complete long pages based on videos, but they do not handle the details of the web pages well, including the corresponding icons and matching product images. A successful reconstruction would require both holistic understanding of the page structure and meticulous attention to its smallest components.

**Case 4 Analysis: Time-Based State Management in a Mobile Viewport.** This case study, classified as [L3, V1, Productivity & Tools], is designed to evaluate a model's ability to handle time-driven state changes, presented within the constraints of a mobile screen resolution. The task is to reconstruct a functional Pomodoro timer. While the visual complexity is low (V1), the mobile viewport requires the model to generate a layout that is responsive or appropriately scaled for a narrow screen. The primary challenge, however, resides in the L3 interaction complexity: the model must implement a state logic governed by both user clicks (e.g., 'start', 'pause') and asynchronous, time-based events (the countdown reaching zero).

As illustrated in Figure 11, GLM-4.5V can successfully implement the interactive operations and logic in the video, but Kimi-VL-thinking is unable to perform subsequent operations because the elements that need to be clicked in the first step are missing in the initial state.

## G   USE OF LARGE LANGUAGE MODELS

We utilized a Large Language Model to assist with grammar correction and language refinement in this paper.

## H   COMPARISON WITH PIPELINE METHODS

As shown in Table 7, a two-stage pipeline method is evaluated to investigate the effect of decoupling perception and generation. In the first stage, the model is prompted to analyze the interaction video and generate a structured description of user actions (the prompt is detailed in Figure 12). In the second stage, this generated description, together with the original visual inputs (video and stitched images), is fed to the model to produce the final code (the prompt is detailed in Figure 13). The results indicate that this approach does not yield significant performance gains. This outcome suggests that the IWR-Bench task requires a tight integration of visual perception, reasoning, and code generation. Critical information, such as assets matching and high-fidelity visual layout, is difficult to fully encapsulate in textual descriptions. Consequently, the two-stage pipeline approach proves less effective for this task.

## I   ANNOTATION GUIDELINES

We have provided the annotation guidelines, as shown in Figure 14 and Figure 15.

Table 7: Performance comparison between End-to-End and Pipeline approaches on IWR-Bench.

| Model | Method | LVS | HVS | VFS | IFS | Final Score |
|---|---|---|---|---|---|---|
| GPT-5 | End-to-End | 68.29 | 60.21 | 64.25 | 24.39 | 36.35 |
| | Pipeline | 67.15 | 58.95 | 63.05 | 25.03 | 36.43 |
| Claude-Sonnet-4 | End-to-End | 65.75 | 56.92 | 61.34 | 22.29 | 34.00 |
| | Pipeline | 64.88 | 58.87 | 61.88 | 23.05 | 34.70 |
| Qwen3-VL (thinking) | End-to-End | 58.55 | 46.13 | 52.34 | 22.07 | 31.15 |
| | Pipeline | 59.21 | 43.58 | 51.39 | 20.31 | 29.64 |
| Qwen2.5-VL-72B | End-to-End | 47.83 | 28.25 | 38.04 | 17.42 | 23.61 |
| | Pipeline | 49.11 | 27.04 | 38.08 | 15.21 | 22.07 |
| Qwen2.5-VL-7B | End-to-End | 28.92 | 12.20 | 20.56 | 13.28 | 15.47 |
| | Pipeline | 26.12 | 12.51 | 19.32 | 12.16 | 14.31 |

## J  TASK SOURCING

Figure 16 displays the examples of our Task Sourcing

## K  RESOLUTION STATISTICS

Table 8 presents the frequency of various resolutions in IWR-Bench.

## L  ASSETS STATISTICS

To investigate the impact of asset quantity on model performance, the distribution of assets across tasks is analyzed and correlated with the performance of the state-of-the-art model, GPT-5. As shown in Table 9, IWR-Bench contains an average of 74.02 assets per task, with a minimum of 1 and a maximum of 502 assets. Figure 17 and Figure 18 summarize the relationship between asset count and model performance. As shown in Figure 18, the Visual Fidelity Score (VFS) tends to decrease as the number of assets increases. Tasks with many assets require more fine-grained visual grounding and precise asset matching.

In contrast to the VFS trend, Figure 17 shows that the overall Final Score exhibits a generally positive correlation with asset count. This effect appears counter-intuitive if assets are considered in isolation and is better explained by the interaction between asset density and task type in IWR-Bench. In practice, many tasks with few assets correspond to high interaction complexity. For example, the 2048 game case contains only a small number of visual assets, yet requires models to reconstruct non-trivial algorithmic, event-driven logic. Conversely, the Apple product introduction page, which includes a rich set of images and icons but is dominated by static content consumption and scrolling.

Table 8: Resolution Statistics

| Resolution | Count |
|---|---|
| 2560x1304 | 40 |
| 1920x990 | 21 |
| 1920x944 | 20 |
| 644x1398 | 8 |
| 430x932 | 4 |
| 1920x924 | 2 |
| 2560x1262 | 2 |
| 2560x1270 | 2 |
| 2560x1260 | 2 |
| 1920x926 | 2 |
| 2560x1286 | 2 |
| 2560x1288 | 2 |
| 2180x1304 | 2 |
| 1920x968 | 1 |
| 1846x886 | 1 |
| 2560x1392 | 1 |
| 1920x922 | 1 |

Table 9: Statistics and distribution of asset counts per task in IWR-Bench.

| Statistic | Value |
|---|---|
| Minimum Assets | 1 |
| Maximum Assets | 502 |
| Average Assets | 74.02 |
| **Asset Count** | **Number of Tasks** |
| 0–39 | 46 |
| 40–79 | 35 |
| 80–119 | 15 |
| $\geq 120$ | 17 |
| *Total Tasks* | *113* |

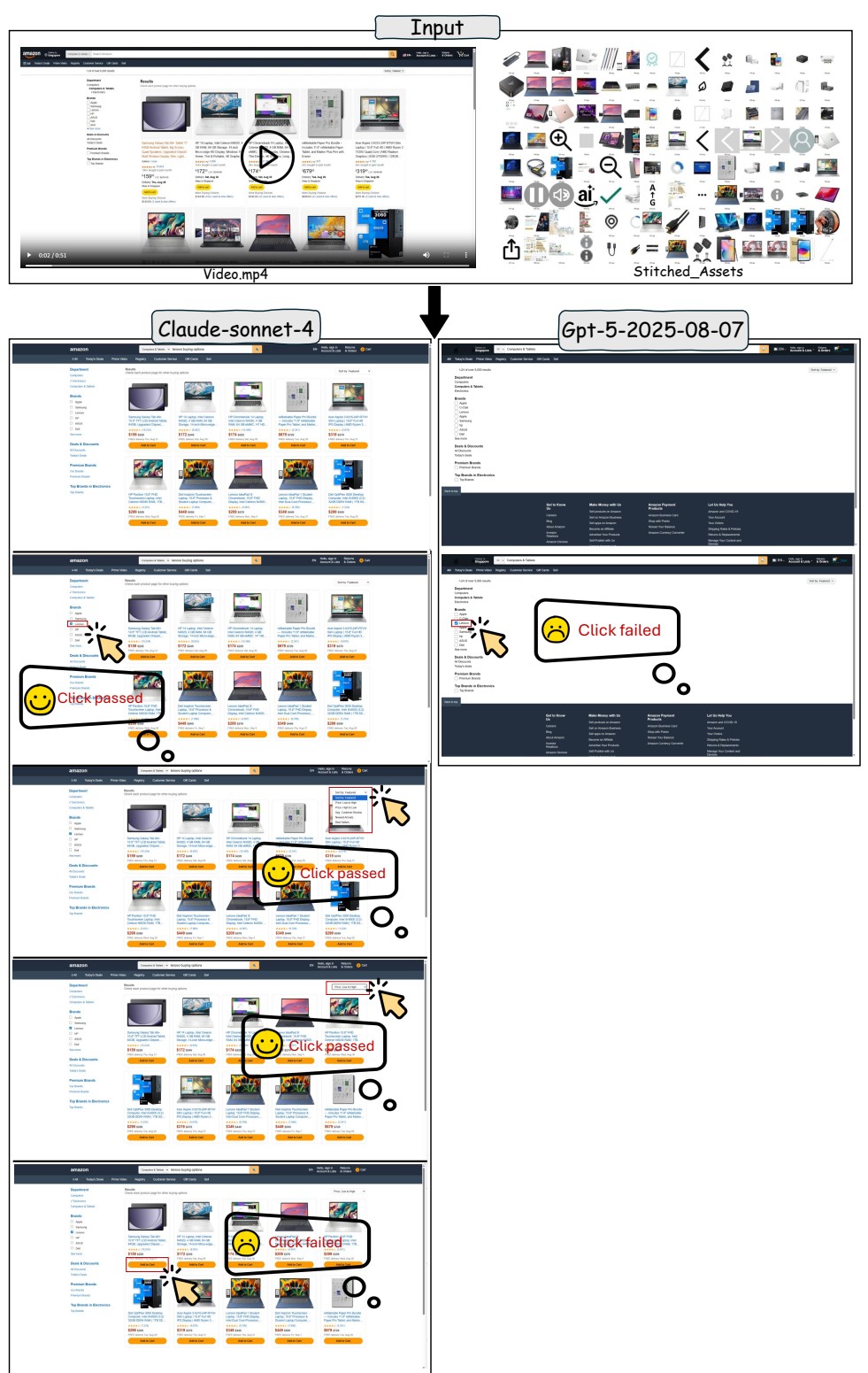

Figure 8: **Case 1: Multi-Step E-commerce Workflow.** This task, classified as [**L2, V2, E-commerce**], requires reconstructing a core e-commerce workflow involving filtering products, sorting the results, and adding an item to the shopping cart.

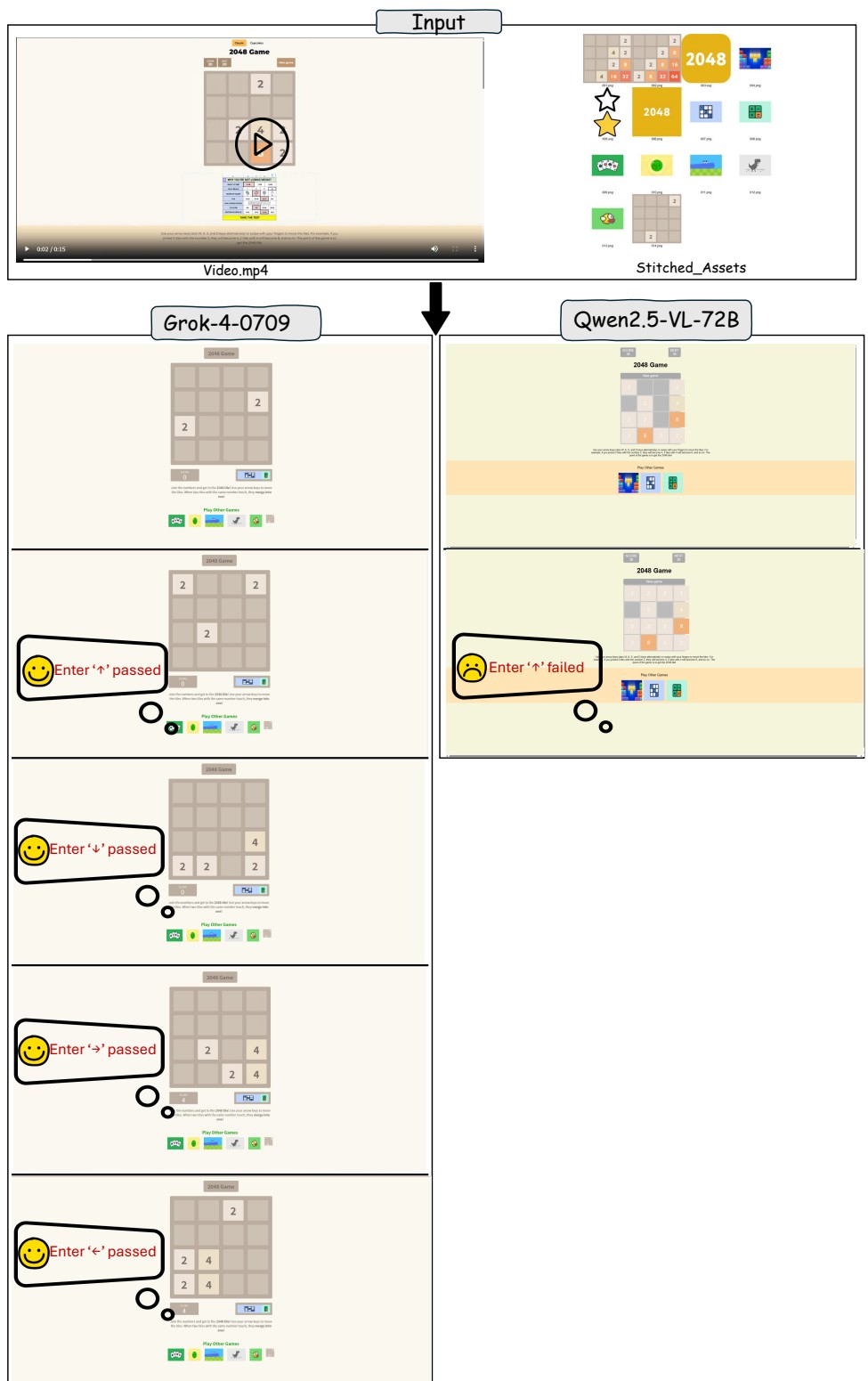

Figure 9: **Case 2: Algorithmic Game Logic Reconstruction.** This task challenges models to reconstruct the rules of the simple browser game 2048. Classified as **[L4, V2, Gaming]**, the primary difficulty lies in algorithmic correctness, not visual complexity.

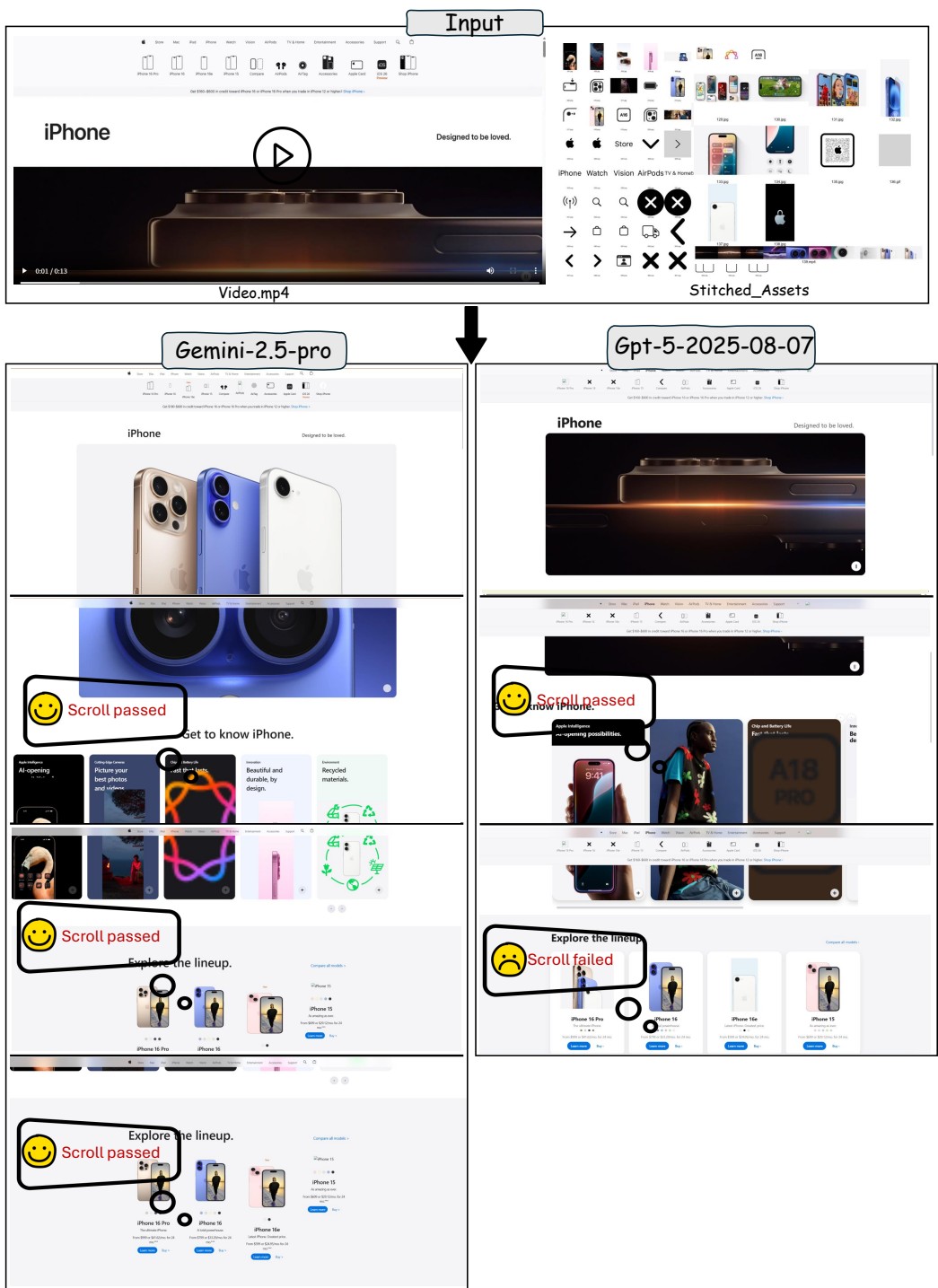

Figure 10: **Case 3: Full-Page Reconstruction with Long Scrolling.** This task focuses on a fundamental capability: reconstructing a webpage that extends far beyond the initial viewport. Classified as **[L1, V3, E-commerce]**, it tests the model's ability to handle static content at scale.

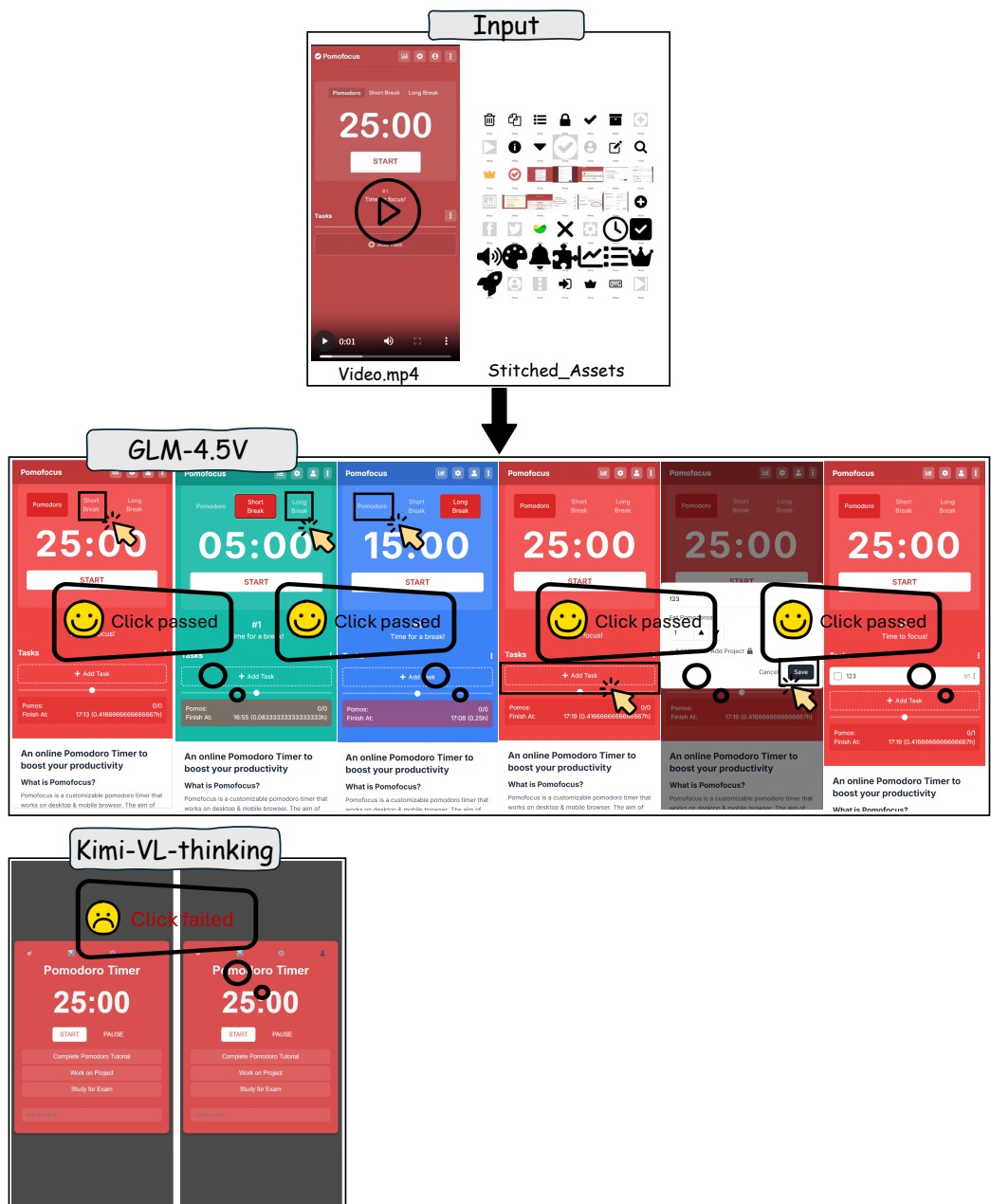

Figure 11: **Case 4: Pomodoro Timer Logic within a Mobile Viewport.** This task requires reconstructing a Pomodoro timer rendered at a mobile resolution. Classified as **[L3, V1, Productivity & Tools]**, the core challenge is not the simple layout but the implementation of time-based state transitions (start, pause, reset).

---

**Prompt template for Stage 1 of the pipeline method.**

Your task is to analyze the provided video frames and describe the user interactions in a structured JSON format.
Your output MUST be a single, valid JSON object.
The JSON should contain an array named 'interactions'. Each object in the array represents one step and must include:
- 'step': (Integer) The action number.
- 'user_action': (String) A clear description of the user's action (e.g., "Click the 'Submit' button").
- 'state_change': (String) A description of the visual changes on the screen after the action.
- 'inferred_logic': (String) The functional rule demonstrated by this interaction (e.g., "Clicking 'Submit' validates the form and shows a success message.").
**Example Object:**
{
"step": 1,
"user_action": "Click the 9th star for rating.",
"state_change": "The first 9 stars turn yellow and a message 'Rating updated: 9/10' appears.",
"inferred_logic": "The webpage should update the rating to the selected star's value and confirm it with a message."
}
Analyze the video frames and generate the complete JSON.

---

Figure 12: Prompt template for Stage 1 of the pipeline method. The model is instructed to analyze the interaction video and generate a structured JSON description containing user actions, state changes, and inferred functional logic for each interaction step.

---

**Prompt template for Stage 2 of the pipeline method.**

You are an expert front-end developer. Your task is to create a pixel-perfect replica of a website from a video and the interactions json.
Generate a single 'index.html' file that contains all HTML, CSS, and JavaScript necessary to replicate the UI, content, and interaction features shown. The webpage resolution in the video is <resolution>.
Instructions:
1. Single File Output: All HTML, CSS, and JS must be in one 'index.html' file.
2. If backend logic is implied, mock it in JS with static data (e.g., a JS array for a fake API call).
3. Assets(Images and Videos in the webpage):
    – All images must use the provided stitched image assets.
    – The 'src' attribute must start with the literal, unchanging string '__PLACEHOLDER_ASSETS_BASE_DIR__/', followed by the actual filename identified from the stitched image.
    – For example: 'src="__PLACEHOLDER_ASSETS_BASE_DIR__/asset001.svg"'.
    – '' tags must include 'width' and 'height' attributes.
    – The provided stitched image assets are before the video.
4. No External Dependencies: The generated code must be entirely self-contained. No External Libraries and no External Fonts.
5. Final Response: Return **only the complete HTML code** in a single "'html code block, with no additional text or explanations.
Interactions JSON: <interactions_json>

---

Figure 13: Prompt template for Stage 2 of the pipeline method. The model receives the structured interaction description from Stage 1, along with the original video frames and extracted assets, to generate the complete webpage code.

**Core Handbook for Data Collection and Annotation**

**I. Core Principles (The Four Cornerstones)**

1. **Clarity**: The task's objective and the user's actions must be unambiguous and instantly understandable.
2. **Controllability**: The recording environment must be strictly controlled to ensure that every task is 100% reproducible.
3. **Evaluability**: Each task must be broken down into a sequence of discrete, verifiable steps and actions.
4. **Consistency**: All annotators must utilize the same tools, settings, and procedures to guarantee dataset consistency.

**II. The Three-Step Recording Process**

**Step 1: Pre-Recording Preparation**

Before initiating a recording, you must complete the following preparatory steps:

1. **Task Familiarization**: Thoroughly understand the task's objective and the required operational flow (e.g., game rules).
2. **Classification Review**: Confirm that the task's [L, V, Domain] classification is appropriate, modifying it with a note if necessary.
3. **Final Checklist (All items must be "Yes")**:

     **[ ] Compliance**:
     - Can the task be completed within a **single** browser tab?
     - Does the task avoid any file uploads?
     - Are all image assets controllable and not dependent on features like infinite scrolling?

     **[ ] Feasibility**:
     - Is the webpage free of sensitive content (political, private)?
     - Does it avoid complex CAPTCHAs (simple numerical recognition is acceptable)?
     - Is the task duration reasonable, ideally between **15 and 200 seconds**?
     - Has the optimal sequence of operations been planned in advance?

**Step 2: Standardized Recording Environment Setup**

To ensure visual consistency, all recordings must be conducted within the following standardized environment:

- **Browser**: Google Chrome (Stable version).
- **Browser State**:
    - Use an **Incognito mode** window for every session.
    - **Disable all browser extensions** to prevent interference.
    - Start a **fresh Incognito window** for each new task.
- **Page Preparation**:
    - If login is required, log in first and **begin recording from the post-login page**.
    - Close any pop-up advertisements whenever possible.
- **Recording Tool & Settings**:
    - Use the officially provided screen recording software.
    - Scope: **Record the browser viewport only**, excluding the address bar, bookmarks, or OS UI.

**Step 3: Normative Recording Procedure**

Execute the recording in a manner that ensures clarity and reproducibility.

- **Starting Point**: Begin recording only after the start URL has fully loaded in the browser.
- **Mouse Operations**: Ensure all movements are **deliberate and purposeful**, following a **Move -> Pause -> Act** pattern, while avoiding meaningless or rapid cursor movements.

Figure 14: Annotation Guidelines 1

- **Pacing**: Wait for all page elements, animations, and transitions to fully load or complete before proceeding to the next action.
- **Integrity**: Record the entire process from the starting point to the final, stable state of task completion.**If any mistake occurs, you must abort the recording and start new.**

**III. Post-Processing: Screenshots and Assertions**

After recording, the video must be processed to add evaluation point screenshots, assertions.

**1. Evaluation Point Screenshots**

**Definition**: Screenshots captured at key moments when the page's visual state undergoes a **significant and stable change**. **Naming Convention**: eval-point-00.png, eval-point-01.png, etc., incrementing sequentially from 00.

**When to Capture a Screenshot:**

- **Capture Is Mandatory**:
  i. **Initial State**: The stable page before any user action is taken.
  ii. **Post-Interaction State**: After an action causes a stable visual change, such as content updating after a filter is applied or a modal window appearing.
  iii. **Final State**: The stable page after the task is fully completed.
- **Capture Is Generally Avoided**:
  - Transient effects from mouse hovers.
  - The process of typing in an input field (only capture after typing is complete *and* triggers a page update).
  - Intermediate frames during a smooth scroll.

**2. Assertion Annotation**

**Definition**: A textual statement used to verify the logical correctness of an action's outcome, especially for non-visual changes like price calculations or sorting logic.

**Principles for Writing Assertions**:

1. **Binary**: The statement must be verifiable as unequivocally true or false.
   - **Good**: "The number in the top-right corner of the cart icon is '3'."
   - **Bad**: "The cart seems to have been updated."
2. **Precise**: The statement must describe specific, quantifiable facts.
   - **Good**: "The text in the 'Total Cost' field reads '$2,049.99'."
   - **Bad**: "The price looks correct."
3. **Targeted**: The statement should focus on logic, data,etc, not purely on visual style.
   - **Good**: "After applying the filter, the result count changed from '100 items' to '15 items'."
   - **Bad**: "After applying the filter, the list's background turned gray."

**When to Add an Assertion:**

- **After List Content Changes**: Following sorting or filtering operations.
  - *Example*: "After sorting by 'Price: Low to High,' the price of the first item should be less than or equal to the second."
- **After Numerical Calculations**: When a shopping cart total price updates.
  - *Example*: "After changing the item quantity from 1 to 2, the total price should update to '€198.00'."
- **After State Changes and Feedback**: When feedback appears after a form submission.
  - *Example*: "After entering an invalid password, an error message 'Incorrect password format' appears."
- **For Game/Algorithm Logic**: Verifying rules in games like 2048 or Sudoku.
  - *Example*: "After merging two '4' tiles, a new '8' tile appears, and the score increases by 8 points."

Figure 15: Annotation Guidelines 2

| Website page URL | Operation Description (High Level) | Complexity of Interaction Logic | Complexity of Layout | Domain | Desktop/ Mobile | Proposer |
|---|---|---|---|---|---|---|
| https://www.apple.com/iphone/ | Scroll down for the iPhone product introduction. | L1 | V3 | Business & Services | Desktop | expert 2 |
| https://www.amazon.com/s?i=specialty-aps&bbn=16225007011&rh=n%3A1622500 7011%2Cn%3A13896617011&ref=nav_em__ nav_desktop_sa_intl_computers_tablets_0_2 _7_4 | On the Amazon page, select "Lenovo" as the brand, sort by price from low to high, then select the lowest-priced physical item and add it to the shopping cart. | L2 | V2 | Business & Services | Desktop | expert 5 |
| https://wandb.ai/zhejiangu/geo3k_async_rl ?nw=nwusershenyfzju | In the W&B experiment log, expand the charts for `training`, `timing_pertoken_ms`, and `response_length` in sequence. Finally, in the `response_length` chart, check the value of `response_length` at the fifth step. | L2 | V4 | Productivi ty & Tools | Desktop | expert 4 |
| https://www.youtube.com/shorts | Swipe down to the third Short, then swipe up to the second Short and open the comment section. | L2 | V2 | Entertain ment & Media | Desktop | expert 3 |
| https://www.hulu.com/welcome | As you browse the homepage, an automatic zoom will occur upon reaching the bottom, and a pop-up will appear after a certain period of time. | L1 | V3 | Entertain ment & Media | Desktop | expert 5 |
| https://minesweeper.online/game/4945019 480 | In the Minesweeper game, click "Beginner" or "Intermediate" at the top to switch between different sizes. Click "Custom" to enter a custom size of 30x20 with 100 mines. Then, start the game. The game will end after 10 clicks or if you hit a mine sooner. | L4 | V2 | Entertain ment & Media | Desktop | expert 2 |
| https://www.nytimes.com/games/wordle/in dex.html | Letter Guessing Game, start the game, an error is prompted when trying to input a non-English word, keep playing until one round is complete. | L4 | V2 | Entertain ment & Media | Desktop | expert 5 |
| https://papergames.io/en/gomoku | Gomoku game: start recording after clicking 'play with robot', and continue for up to 30 clicks or until an early win or loss. | L4 | V2 | Entertain ment & Media | Desktop | expert 1 |
| https://2048.gg/en | In the 2048 game, press the up, down, left, and right arrow keys in sequence. | L4 | V2 | Entertain ment & Media | Desktop | expert 2 |
| https://www.imdb.com/chart/top/ | Browse the top 50 movies from the IMDb Top 250. | L1 | V2 | Entertain ment & Media | Desktop | expert 2 |
| https://www.imdb.com/ | On the IMDb homepage, search for "The Dark Knight". Once on the movie's main page, scroll down and click the "+Review" button. Select a 9-star rating, enter "good movie" for the title, and submit the review. | L3 | V3 | Entertain ment & Media | Desktop | expert 4 |
| https://music.apple.com/en/new | On the Apple Music home page, click to play the first of the latest songs, then click the player bar at the top to enter full screen, and then click the pause button to pause playback. | L3 | V3 | Entertain ment & Media | Desktop | expert 3 |
| https://www.goodreads.com/ | On the homepage, scroll down to find "Readers' Favorite 2024" from "Goodreads Choice Awards: Readers' Favorite Books 2024". Once the new page opens, find "2022 Awards" on the left. | L2 | V2 | Entertain ment & Media | Desktop | expert 4 |
| https://www.wikipedia.org/ | Search for "artificial intelligence". After the page opens, click the "Future" button in the directory on the left to go to the Future page, then click the "Goals" button on the left to go to the Goals page. | L2 | V1 | Entertain ment & Media | Desktop | expert 1 |
| https://www.nytimes.com/ | Select "Your Money" under "Business" at the top, and then click on an article to browse. | L2 | V1 | Entertain ment & Media | Desktop | expert 5 |
| https://store.steampowered.com/ | Open the Steam homepage, search for 2077, and open its game page. Verify your age by entering 1975 to access the main game page. Click 'Add to Cart', and a page will appear confirming it has been successfully added to your cart. | L2 | V4 | Entertain ment & Media | Desktop | expert 4 |
| https://store.steampowered.com/charts/ | Open the page, scroll down and click "see all 100 top sells". Find the list, click on the first three discounted items and add them to the cart. Then, open the shopping cart and check the three items. | L3 | V4 | Entertain ment & Media | Desktop | expert 4 |
| https://www.polygon.com/ | Open the homepage and swipe down to browse information. | L1 | V3 | Entertain ment & Media | Desktop | expert 2 |
| https://papergames.io/en/chess | Chess game: Start recording after clicking 'Play with Robot' to begin. The recording stops after a maximum of 30 moves or if the game is won or lost sooner. | L4 | V2 | Entertain ment & Media | Desktop | expert 4 |
| https://papergames.io/en/tic-tac-toe | Tic-Tac-Toe Game: Start recording after clicking 'play with robot'. The recording will end after a maximum of 10 clicks or when the game is won or lost. | L4 | V2 | Entertain ment & Media | Desktop | expert 5 |
| https://www.ebay.com/ | Browse the homepage and favorite three different items. | L2 | V2 | Business & Services | Desktop | expert 3 |
| https://www.walmart.com/ip/Hawaiian-Tropic-Sheer-Touch-Ultra-Radiance-50-SPF-Adult-Sunscreen-Lotion-8-fl-oz/15610900 | Switch the Pack Size from single to 2-pack, and then to 3-pack. | L2 | V3 | Business & Services | Desktop | expert 3 |

Figure 16: Examples of Task Sourcing

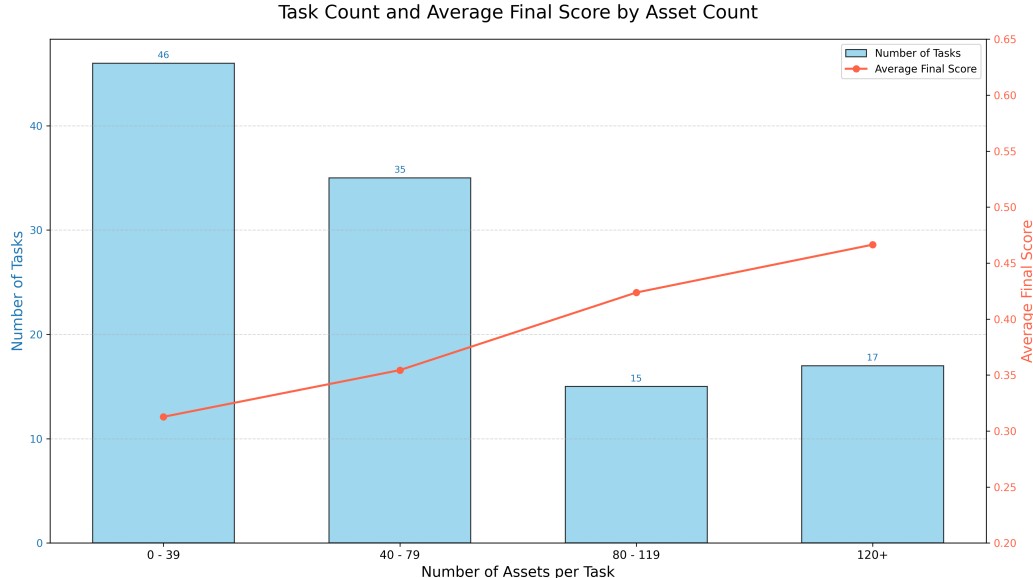

Figure 17: Task Count and Average final score by Asset Count

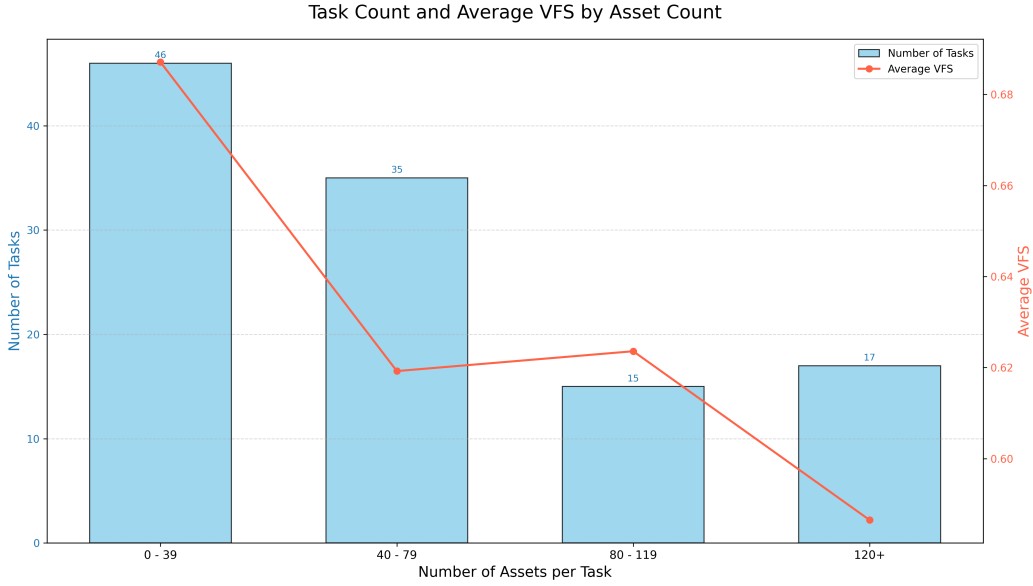

Figure 18: Task Count and Average VFS by Asset Count

