# OpenReview forum: "IWR-Bench: Can LVLMs reconstruct interactive webpage from a user interaction video?"
_ICLR.cc/2026/Conference — ICLR 2026 Poster_

### Official Review · Reviewer_dLFH · 2025-10-30

**Soundness:** 3
**Presentation:** 3
**Contribution:** 3
**Rating:** 4
**Confidence:** 4

**Summary:**

The paper shifts evaluation from static mimicry to working webpage from interaction videos, revealing a wide functionality gap in today’s LVLMs and offering a rigorous, automated protocol to measure progress.

**Strengths:**

1. Shifts the problem from static screenshot‑to‑code to interactive webpage reconstruction from video;
2. First benchmark to combine videos and ground‑truth action trajectories for web reconstruction, filling the gap between static page benchmarks and general video QA.

**Weaknesses:**

1. With only 113 tasks, the benchmark is relatively small compared to other vision-language benchmarks;
2. The authors set w=0.5 for VFS and α=0.7 for the Final Score “based on validation studies,” but details are limited in the paper;
3. A single MLLM (Gemini‑2.5‑Pro) is used in both logical asserts verification and providing the visual score, which risks bias and errors.

**Questions:**

1. Task source: the paper mentions "Experts in web development", diversity balancing, and the three-step annotation. However, there is no any detail, such as concrete criteria and inter-annotator agreement. Would the authors supply more details?
2. VFS: VFS is computed only on checkpoints that are both reached and flagged, so that the missing states never contribute. Models that fail early avoid penalties on hard visual states, with higher VFS. please clarify the metric in details.
3. Weights: VFS blends the scores with w=0.5; the Final Score uses α=0.7. The paper cites a validation study but provides few in‑text sensitivity analyses.
4. Single‑judge bias: The same MLLM (Gemini‑2.5‑Pro) is used to verify logical assertions and provide the high‑level visual score, and this model also appears on the leaderboard. Were the results verified by humans? i am concern its reproducibility and stability. A single MLLM pass determines assertions, which may exhibits sampling variance. The authors should run multiple independent judging passes and aggregate the results.

---

> ### Author Response · Authors · 2025-11-19
> **Response [1/3]**
>
> ### [Concern 1 (Weakness 1)] With only 113 tasks, the benchmark is relatively small
>
>
> Thank you for your valuable feedback regarding the benchmark's scale. For clarification and to provide context, the number of tasks in IWR-Bench is comparable to established benchmarks in this domain. For instance, Interaction2Code[1] comprises 127 webpages, Design2Code[2] includes 484 screenshots (but only 80 in Design2Code-HARD[2]), and CC-HARD contains 128 screenshots. The design of IWR-Bench prioritizes this principle of depth and complexity. The 113 tasks are intentionally granular, comprising 1,001 distinct actions, 620 visual checkpoints, and 403 logical assertions. This granular structure is essential for rigorously evaluating the complex, stateful workflows that define this new task. Therefore, the action-level sample size of **1,001** serves as a more representative measure of the benchmark's scale and challenge.
>
> Furthermore, broad coverage is systematically ensured. Tasks are taxonomized across three orthogonal axes [Domain(5 major categories with 16 subcategories), Visual Complexity(V1/V2/V3/V4 = 11.50%/43.36%/35.40%/9.73%), Interaction Logic((L1/L2/L3/L4 = 7.96%/47.79%/33.63%/10.62%))] and multiple resolutions to cover a wide spectrum of real-world web scenarios. This curation strategy establishes a high-quality, high-complexity benchmark designed for comprehensive evaluation.
>
>
> [1]Xiao, Jingyu, et al. "Interaction2Code: Benchmarking MLLM-based Interactive Webpage Code Generation from Interactive Prototyping." arXiv preprint arXiv:2411.03292 (2024).
>
> [2]Si, Chenglei, et al. "Design2code: Benchmarking multimodal code generation for automated front-end engineering." Proceedings of the 2025 Conference of the Nations of the Americas Chapter of the Association for Computational Linguistics. 2025.
>
> [3]Gui, Yi, et al. "Latcoder: Converting webpage design to code with layout-as-thought." Proceedings of the 31st ACM SIGKDD Conference on Knowledge Discovery and Data Mining V. 2. 2025.

---

> ### Author Response · Authors · 2025-11-19
> **Response  [2/3]**
>
> ### [Concern 2 (Question 1)] Task source
>
> The annotation process description in Section 3.2 may benefit from additional clarification. To provide more comprehensive details, we have updated the paper with two new appendices in the updated version.
>
> The **APPENDIX I ANNOTATION GUIDELINES** provides the comprehensive annotation manual given to the annotators. This manual details the concrete criteria for the three-step annotation process, specifying how to author natural language descriptions, assign visual evaluation flags, and define logical assertions.
>
> The **APPENDIX  J TASK SOURCING** presents the initial pool of tasks sourced by "Experts in web development", illustrating their professional criteria for identifying common user goals and diverse web functionalities.
>
> Regarding inter-annotator agreement, it is acknowledged that IAA metric is standard for large-scale, repetitive annotation tasks. However, for our complex benchmark, IWR-Bench requires a complex annotation process and multi-dimensional annotation acceptance, so a more rigorous, iterative expert-led verification process was adopted. Instead of just measuring agreement on a static set of guidelines, our process involved an author-led verification loop. As described in "Verification and Finalization"  of Figure 4, any task failing the authors' expert inspection was returned for revision. This iterative refinement process continued until every task achieved a standard of correctness and clarity deemed necessary for this new, challenging problem. This protocol prioritizes the **correctness and unambiguity** of every single task, a standard that is arguably more suitable for a foundational benchmark of this nature.

---

> ### Author Response · Authors · 2025-11-19
> **Response [3/3]**
>
> ### [Concern 3 (Question 2)] VFS
>
> The observation that VFS is computed only on reached checkpoints is correct. This is a deliberate design choice to isolate visual fidelity evaluation from functional correctness. VFS is designed to answer a specific question: "For the parts of the webpage that the model *successfully* generated and reached, how visually accurate are they?" It is an independent measure of rendering quality. This is a deliberate choice, as the complexity of subsequent pages in a web navigation task is not consistently higher or lower. Therefore, for models that fail early to avoid penalties on hard visual states, there is not necessarily higher VFS. And the concern that early failure may artificially inflate VFS is mitigated at the Final Score level. Early failure drives the IFS towards zero, and the Final Score would be lower.
>
>
> ---
>
> ### [Concern 4 (Weakness2 & Question 3)] Weights
>
> Thank you very much for your valuable feedback. The detailed analysis is provided in Appendix C, with a summary presented here for your convenience.
>
> To ground our metrics in human perception, a human alignment study was conducted. This study involved 5 PhD-level students who rated 60 outputs, sampled from various models, on two criteria: visual fidelity and overall quality. A grid search over the set {0.1, ..., 0.9} was then performed to find the weights `w` and `α` that maximize the Spearman's ρ correlation between our automated scores and these human judgments.
>
> The analysis reveals that the peak correlation for VFS is achieved at `w=0.5` (ρ=0.57) and for the Final Score at `α=0.7` (ρ=0.65). This data-driven approach provides a principled and reproducible method for metric parameterization. Unlike prior work that often reports multiple, unweighted scores, our unified Final Score offers a holistic and human-aligned summary of a model's performance.
>
> ---
>
> ### [Concern 5 (Weakness3 & Question 4)] Single‑judge bias
>
> We would like to clarify that the evaluation results were rigorously validated through both human oversight and quantitative analysis to ensure fairness and reliability.
>
> **Human Verification:** The benchmark’s integrity is underpinned by human verification at multiple stages. As detailed in Section 5.4, the agent-as-a-judge framework itself was validated by manual inspection. Three PhD students observed the agent's execution on 100 randomly sampled webpages to confirm its operational fidelity. This verification process specifically included scrutinizing the MLLM's judgments on logical assertions against the observed browser behavior. Crucially, not a single error from an incorrect logical assertion judgment was identified across all 100 inspected cases.
>
> **Reliability of the MLLM Judge:** The reliability of using Gemini-2.5-Pro as a judge is supported by strong empirical evidence. For visual scoring, the High-level Visual Score (HVS) generated by the MLLM judge shows a Pearson correlation of **0.9806** (Calculated using the main evaluation data from the original paper) with the objective Low-level Visual Score (LVS), demonstrating alignment. For logical assertions, the reliability stems from their deliberate design. Assertions are intentionally **atomic and verifiable** to minimize judgment complexity. For instance, to verify a 'sort by price' action, the assertion is not a holistic 'check if all items are sorted,' which could introduce ambiguity. Instead, it is a concrete, localized check like **'verify that the price of the first item is lower than the second.'** This design reduces the verification to a direct comparison, significantly mitigating the potential for judgment bias. Furthermore, the ground-truth logical assertions themselves were cross-verified by human annotators during the benchmark construction phase (Section 3.2), ensuring the correctness of the evaluation standard.
>
>
> **On Using the Same MLLM as Judge and Competitor:** Using the most capable model as a judge is a standard and necessary practice in the "LLM-as-a-judge" paradigm[1] to ensure the highest quality of automated evaluation. The key is to ensure the judging task is distinct from the generation task. In our case, the generation task is a complex, creative synthesis of code from video. The judging task, by contrast, is a constrained, analytical verification of atomic facts. The different nature of these tasks mitigates the risk of self-serving bias.
>
>
> [1]Zheng, Lianmin, et al. "Judging llm-as-a-judge with mt-bench and chatbot arena." Advances in neural information processing systems 36 (2023): 46595-46623.
>
>
> ---
> Once again, we thank you for your constructive comments, which have helped us significantly improve the paper. All newly added content has been incorporated into the updated revision and highlighted in RED.

---

> ### Author Response · Authors · 2025-11-26
> **Short Summary of Discussion**
>
> Dear Reviewer dLFH, thank you for your rigorous review of our benchmark design and evaluation protocol. As the rebuttal window is closing and it has been about a week since we submitted our detailed reply, we kindly ask you to consider the following clarifications:
>
> (1) On benchmark scale and sourcing, we emphasized that the 113 tasks expand to 1,001 actions and added two appendices detailing expert task sourcing and full annotation guidelines to make the process transparent.
>
> (2) For VFS and the weighting scheme, we clarified that VFS is intentionally defined on reached checkpoints to isolate rendering quality, and that the weights are chosen via a human alignment study that maximizes Spearman correlation with expert judgments.
>
> (3) Regarding single-judge bias, we reported human validation of the Gemini-2.5-Pro judge  and a strong correlation between its high-level visual scores and objective low-level metrics, which together support the reliability and reproducibility of our agent-as-a-judge setup.
>
> These methodological details and validations have been incorporated into the revised version (highlighted in red). **We would greatly appreciate it if you could let us know whether these clarifications satisfactorily address your concerns**, and we are very willing to further elaborate on any point that may help you finalize a fair and confident evaluation.

---

### Official Review · Reviewer_czER · 2025-10-31

**Soundness:** 4
**Presentation:** 4
**Contribution:** 3
**Rating:** 6
**Confidence:** 4

**Summary:**

The authors introduce the benchmark IWR-Bench, a benchmark for video to code generation. In this benchmark, the multimodal large language model (MLLM) receives a video of a user interacting with a website, along with anonymized assets, and must generate code that reproduces the same functionality. They also propose a framework for evaluating the generated code using two metrics: the Interaction Functionality Score (IFS) for functional accuracy and the Visual Fidelity Score (VFS) for visual correctness. The overall performance is reported as a weighted combination of these two metrics. The authors evaluate 28 MLLMs, including GPT-5, Sonnet-4, and Gemini-2.5-Pro, showing that current models still struggle with this challenging task.

**Strengths:**

- The paper defines a novel and interesting task, video-2-code, which is both challenging and potentially useful for practical applications.

- The paper is well-written and provides detailed descriptions of data generation, evaluation metrics, and experimental setup.

**Weaknesses:**

- Missing citation: the benchmark PairBench [1] investigates when an MLLM can act as a reliable judge and could be used to identify an agent-as-a-judge suitable for this task.

- Minor typos, such as inverted quotation marks.

- The paper does not report the average, maximum, and minimum number of assets per task, which could help analyze how performance varies with the number of input assets. A results table showing model performance grouped by asset count would make this analysis clearer and more informative.

[1] PairBench: Are Vision-Language Models Reliable at Comparing What They See?

**Questions:**

- What are the resolutions of the videos used in the benchmark?

- How much does it cost to run the benchmark on proprietary models versus open-source models, given that the agent-as-a-LLM is Gemini-2.5-Pro? This would help assess the practicality of the benchmark for everyday use.

---

> ### Author Response · Authors · 2025-11-19
> **Response**
>
> Thank you for the constructive feedback and the acknowledgment of the task's novelty. Reponses to specific concerns are provided below.
>
>
> ### [Concern 1 (Weakness 1)] Missing citation
>
> The citation for PairBench [1] is incorporated into the "LVLM Benchmarks" section. It provides valuable context for the "agent-as-a-judge" framework employed in this benchmark.
>
> ### [Concern 2 (Weakness 2)] Minor typos
>
> The manuscript is thoroughly proofread. Typographical errors, including the inverted quotation marks, are corrected in the revised version.
>
> ### [Concern 3 (Weakness 3)] Number of assets
>
> Statistics regarding the number of assets are added to Table 9 in the **Appendix L ASSETS STATISTICS**(using red text). The average, maximum, and minimum asset counts are **74.02**, **502**, and **1**.
>
> To analyze how performance varies with asset count, results grouped by asset quantity are provided. The trends are visualized in Figure 17 and Figure 18. A decrease in Visual Fidelity Score (VFS) is observed as the asset count increases, which reflects the difficulty of fine-grained visual grounding in asset-heavy cases. In contrast, the Final Score tends to improve with more assets. This pattern is linked to the task design in IWR-Bench, where low-asset tasks often involve more complex algorithmic logic (for example, the *2048* game). This suggestion helps reveal a trade-off between visual complexity and logical complexity.
>
> ### [Concern 4 (Question 1)] Resolutions of the videos
>
> Video resolution statistics are added to **Appendix K. Resolution Statistics** (Table 8, highlighted in red). The dataset covers a diverse range of resolutions, with **2560x1304** and **1920x990** being the most frequent. This diversity stems from the native screen specifications of various devices (e.g., mobile and desktop interfaces) used during data collection, ensuring robustness to different aspect ratios.
>
> ### [Concern 5 (Question 2)] Cost estimation
>
> The cost of running the benchmark is estimated based on Google Gemini Developer Batch API pricing for the evaluator agent (Gemini-2.5-Pro) [2]. The evaluation involves three types of calls:
>
> 1.  Agent Execution: Performing interactions (only click and type need call llm). (~650 input / ~542 output tokens per step).
> 2.  Visual Verification (HVS): Assessing visual fidelity. (~734 input / ~2447 output tokens per check).
> 3.  Logical Assertion: Checking logicalassertions. (~589 input / ~1142 output tokens per check).
>
> Since execution and verification are **only invoked when the page can be reached**, the cost is dynamic:
>
> 1.  Theoretical Maximum: For a hypothetical model that perfectly reproduces all tasks (triggering 777 executions, 620 visual checks, and 403 logical checks), the total benchmark cost is approximately **$12.74**.
> 2.  Actual Scenario: Costs are proportional to the success rate.
>     *   For a SOTA proprietary model (e.g., GPT-5, 36.35% success), the cost is around **$4.63**.
>     *   For an open-source model (e.g., Qwen2.5-VL-7B, 15.47% success), the cost is only **$1.97**.
>
> This analysis demonstrates that the benchmark is inclusive and cost-effective for standard research usage.
>
>
>
> [1] PairBench: Are Vision-Language Models Reliable at Comparing What They See?
>
> [2] Gemini Developer API pricing https://ai.google.dev/gemini-api/docs/pricing#batch_1
>
>
>
>
> ---
> Once again, we thank you for your constructive comments, which have helped us significantly improve the paper. All newly added content has been incorporated into the updated revision and highlighted in RED.

---

> ### Author Response · Authors · 2025-11-26
> **Short Summary of Discussion**
>
> Dear Reviewer czER, thank you for your helpful suggestions to improve completeness. As the discussion phase is ending and our detailed response has been available for about a week, we kindly invite you to review the updates we made in response to your comments:
>
> (1) We added the missing PairBench citation and carefully corrected all typographical issues, including inverted quotation marks.
>
> (2) We provided detailed asset statistics (average 74.02, max 502, min 1 per task) and analyzed performance grouped by asset count, revealing a trade-off between visual complexity and logical complexity.
>
> (3) We reported the video resolution distribution and a concrete cost estimation for running the benchmark with Gemini-2.5-Pro, showing that the evaluation remains practical and affordable for both proprietary and open-source models.
>
> All of these additions are now integrated into the manuscript (highlighted in red). **We would be very thankful if you could confirm that these changes resolve your concerns, or let us know if there is anything else we can clarify**.

---

> > ### Comment · Reviewer_czER · 2025-11-27
> > **Response to Rebuttal**
> >
> > Thank you for providing the additional information and making the edits. I believe the 'PairBench' citation may be problematic, as it currently reads '(SEE, 2025),' while this is the correct paper: [https://arxiv.org/abs/2502.15210]. Please look into fixing the citations. Once addressed, I am happy to increase my score.

---

> > > ### Author Response · Authors · 2025-11-28
> > > **Response**
> > >
> > > Dear Reviewer czER,
> > > Thank you for your careful check and for pointing out the citation error.
> > >
> > > We have corrected the PairBench citation. The reference now properly points to the correct paper and appears accurately in both the in-text citation and the reference list.
> > >
> > > We appreciate your careful review and look forward to your updated assessment.

---

### Official Review · Reviewer_vw9e · 2025-11-01

**Soundness:** 3
**Presentation:** 3
**Contribution:** 2
**Rating:** 4
**Confidence:** 4

**Summary:**

* The paper introduces IWR-Bench, a new benchmark for evaluating vision-language models on the task of reconstructing interactive webpages from videos of user interactions. The benchmark contains 113 tasks collected from real websites, covering 1,001 user actions with diverse layouts, domains, and interaction complexities. Each task includes a user interaction video, crawled static assets, action trajectories, and checkpoint screenshots.
* The valuation is performed using an agent-as-judge method. Twenty-eight LVLMs are included for the evaluation: GPT-5, Claude, Gemini, etc and the top-performing model only obtain a score of 36.35.

**Strengths:**

* The paper provides a comprehensive and well-structured dataset sourced from real-world websites, ensuring realism and diversity across domains, visual styles, and interaction types.
* Extensive empirical study across 28 LVLMs offers valuable baseline results and comparative insights.

**Weaknesses:**

* The difference between this paper and prior work such as Interaction2Code is not substantial. The main distinction lies in the number of interaction steps: Interaction2Code handles single-step interactions, whereas IWR-Bench introduces multi-step sequences. This difference, while incremental, limits the benchmark’s overall novelty and impact.

* For experiments, as mentioned around Line 090, the challenges of interactive webpage reconstruction lie in two aspects: multi-modal perception and code generation. It would be valuable to include pipeline-based baselines. A natural idea would be to use a vision-language model to describe each step of the interaction video in terms of states and action assertions, and then feed this detailed description as a caption to a code-specialized model to generate the webpage code.

**Questions:**

* What is the correlation coefficient between the metric results on the benchmark proposed in this paper and the metric results on Interaction2Code?

---

> ### Author Response · Authors · 2025-11-19
> **Response [1/3]**
>
> ### [Concern 1 (Weakness 1)]  Distinction from Interaction2Code
>
> We sincerely thank you for your insightful feedback. Table 2 in the original paper provides a comparison of the differences at the high-level. The detailed distinction is clarified and strengthened in the revision (using red color text). We appreciate the opportunity to clarify the fundamental differences between IWR-Bench and Interaction2Code.
>
>
> | Aspect | IWR-Bench (Ours) | Interaction2Code | Explanation of the Difference |
> | :--- | :--- | :--- | :--- |
> | **Core Task & Input** | `Video-to-code`  for stateful workflows. | `Image-pair-to-code` for stateless events. | IWR-Bench's video input requires temporal reasoning over event sequences and state transitions. In contrast, Interaction2Code focuses on stateless events captured by two static images. |
> | **Data Source & Complexity** | Tasks from live, contemporary websites. | Tasks from archived C4/GitHub pages. | Tasks are sourced from modern web applications, presenting a more challenging testbed. A three-tiered taxonomy further ensures task complexity and diversity. |
> | **Assets & Realism** | All original static assets (images, icons) are provided. | All static assets are removed from webpages. | The inclusion of all assets necessitates visual-to-asset grounding. Interaction2Code simplifies the task by removing assets, reducing realism. |
> | **Evaluation Method** | Multi-step trajectory execution via an automated agent. | Single-step functional event verification. | The evaluation protocol assesses the functional integrity of the entire user workflow. |
> | **Problem Frontier** | A largely unsolved problem (GPT-5 36.35). | A well-addressed problem for SOTA models(GPT-5 84.65/80.70/71.55). | The significant performance gap on IWR-Bench empirically demonstrates that it establishes a new and more difficult problem frontier for LVLMs. |
>
> **In summary**: IWR-Bench redefines the problem by shifting from **static image pairs** to **dynamic video**, from **archived pages** to **live websites**, and from **asset-free** reconstruction to **asset-complete** synthesis. Current SOTA models (GPT-5) achieve an average score of **80.70** on Interaction2Code but plummet to **36.35** on IWR-Bench. This significant performance drop confirms that IWR-Bench poses new challenges—specifically temporal consistency and asset grounding—that are not covered by prior benchmarks.

---

> ### Author Response · Authors · 2025-11-19
> **Response [2/3]**
>
> ### [Concern 2 (Weakness 2)] Pipeline-based baselines
>
>
> Thank you for your insightful suggestion regarding the inclusion of pipeline-based baselines. This valuable suggestion has been incorporated to strengthen the experimental analysis. An evaluation of a two-stage pipeline approach is now included in **Appendix H** of the updated revision.
>
> Additionally, it is worth clarifying a nuance regarding the challenges mentioned in Line 090. The paper highlights the dual challenges of multi-modal perception **and reasoning**. This reasoning is not limited to logical inference but also encompasses crucial vision-centric aspects, such as the spatial matching of static assets to their dynamic context within the video. This type of visual grounding is difficult to fully capture and convey through intermediate textual descriptions alone. The results from the new experiment support this perspective.
>
> The results (Table 7 in revision) show that **pipeline methods yield no significant performance gain**. In the first stage, the model is prompted to analyze the interaction video and generate a structured description of user actions (the prompt is detailed in Figure 12，updated revision). In the second stage, this generated description, together with the original visual inputs (video and stitched images), is fed to the model to produce the final code (the prompt is detailed in Figure 13, updated revision). The results indicate that this approach does not yield significant performance gains. This outcome suggests that the IWR-Bench task requires a tight integration of visual perception, reasoning, and code generation. Critical information, such as assets matching and high-fidelity visual layout, is difficult to fully encapsulate in textual descriptions.
>
> | **Model**                | **Method** | **LVS** | **HVS** | **VFS** | **IFS** | **Final Score** |
> | ------------------------ | ---------- | ------- | ------- | ------- | ------- | --------------- |
> | **GPT-5**                | End-to-End | 68.29   | 60.21   | 64.25   | 24.39   | 36.35           |
> |                          | Pipeline   | 67.15   | 58.95   | 63.05   | 25.03   | 36.43           |
> | **Claude-Sonnet-4**      | End-to-End | 65.75   | 56.92   | 61.34   | 22.29   | 34.00           |
> |                          | Pipeline   | 64.88   | 58.87   | 61.88   | 23.05   | 34.70           |
> | **Qwen3-VL (thinking)**  | End-to-End | 58.55   | 46.13   | 52.34   | 22.07   | 31.15           |
> |                          | Pipeline   | 59.21   | 43.58   | 51.39   | 20.31   | 29.64           |
> | **Qwen2.5-VL-72B**       | End-to-End | 47.83   | 28.25   | 38.04   | 17.42   | 23.61           |
> |                          | Pipeline   | 49.11   | 27.04   | 38.08   | 15.21   | 22.07           |
> | **Qwen2.5-VL-7B**        | End-to-End | 28.92   | 12.20   | 20.56   | 13.28   | 15.47           |
> |                          | Pipeline   | 26.12   | 12.51   | 19.32   | 12.16   | 14.31           |
>
>
> *Table: Performance comparison between End-to-End and Pipeline approaches on IWR-Bench.*

---

> ### Author Response · Authors · 2025-11-19
> **Response [3/3]**
>
> ### [Concern 3 (Question 1)] Correlation with Interaction2Code
>
>
> A correlation analysis was performed between the final scores on IWR-Bench and the full-page level metrics on Interaction2Code for nine representative models . The Spearman's rank correlation coefficients (ρ) are presented below.
>
>
> | Model                     | Clip Score | Text Score | Struct Score | Interaction2code Avg | IWR-Bench Score | Performance GAP vs IWR-Bench |
> | :------------------------ | :--------: | :--------: | :----------: | :------------------: | :-------------: | :---------------------------: |
> | GPT-5                     |   84.65    |   80.70    |    71.55     |        78.96         |      36.35      |            -42.61             |
> | Claude-Sonnet-4           |   81.87    |   74.91    |    67.72     |        74.83         |      34.00      |            -40.83             |
> | GPT-4.1                   |   79.63    |   67.27    |    65.30     |        70.73         |      31.99      |            -38.74             |
> | Qwen3-VL (thinking)       |   64.99    |   53.69    |    50.35     |        56.34         |      31.15      |            -25.19             |
> | Gemini-2.5-Pro (thinking) |   76.51    |   68.69    |    62.26     |        69.15         |      30.36      |            -38.79             |
> | Gemini-2.5-Pro            |   79.96    |   72.46    |    64.38     |        72.27         |      30.31      |            -41.96             |
> | GPT-4o                    |   71.14    |   51.47    |    52.77     |        58.46         |      29.55      |            -28.91             |
> | GPT-5-nano                |   77.37    |   63.12    |    63.79     |        68.10         |      26.10      |            -42.00             |
> | Qwen2.5-VL-72B            |   64.30    |   41.97    |    42.34     |        49.54         |      23.61      |            -25.93             |
>
>
> | Interaction2Code Metric | Spearman's rho (ρ) with IWR-Bench Score | p-value |
> | :--- | :---: | :---: |
> | Clip Score | 0.7000 | 0.0358 |
> | Text Score | 0.7667 | 0.0159 |
> | Struct Score | 0.7500 | 0.019|
>
>
>
> **Critical Insight on Contribution:**
> While the ranking correlation is moderate (ρ~0.75), the **absolute performance gap is substantial**.
> *   **Performance Drop:** Models show a drastic score decrease on IWR-Bench compared to Interaction2Code (e.g., GPT-5 drops from **78.96** to **36.35**).
> *   **Conclusion:** The moderate correlation confirms validity (it measures relevant skills), but the lack of a perfect correlation (<0.8) and the massive performance drop prove that IWR-Bench introduces **distinct, uncorrelated dimensions of difficulty** (specifically temporal reasoning and complex asset grounding). This evidence supports IWR-Bench as a necessary, complementary benchmark for the next generation of LVLMs.
>
>
>
>
> ---
>
> Once again, we thank you for your constructive comments, which have helped us significantly improve the paper. All newly added content has been incorporated into the updated revision and highlighted in RED.

---

> ### Author Response · Authors · 2025-11-25
> **Additional Response**
>
> ## [Additional Response] Visual Comparison of Dataset Inputs
>
> To further illustrate the fundamental differences between IWR-Bench and Interaction2Code, we have prepared a side-by-side visualization page for you to inspect the actual inputs.
>
> **Anonymous Visualization Link:** https://anonymous.4open.science/w/casecompare_temp-4B4C/
>
> The visualization directly demonstrates two key distinctions:
>
> 1.  **Input Nature:** IWR-Bench uses **continuous interaction videos** paired with **raw static assets**, whereas Interaction2Code relies on discrete, static image pairs. This necessitates temporal reasoning and visual-asset grounding in IWR-Bench.
>
> 2.  **Complexity & Diversity:** IWR-Bench samples are sourced from modern, live websites featuring high-fidelity layouts and diverse visual styles, presenting a more challenging distribution compared to prior benchmarks.
>
> We hope this visual evidence helps you confirm that IWR-Bench presents a fundamentally different and more challenging problem setting. We remain available for any further discussion.

---

> ### Author Response · Authors · 2025-11-26
> **Short Summary of Discussion**
>
> Dear Reviewer vw9e, Thank you for your insightful suggestions. As the discussion period is coming to a close and our detailed response has been posted for a week, we kindly ask you to review our updates addressing your concerns:
>
> (1) **Distinction from Interaction2Code:** We detailed differences across 5 dimensions (e.g., Core Task, Input, Realism). This distinguishes IWR-Bench as a **"novel and interesting task"** (Reviewer 7Tx2, Reviewer czER) that **"shifts the problem from static screenshot-to-code to interactive webpage reconstruction"** (Reviewer dLFH). To intuitively demonstrate this, we also provided a **side-by-side visualization page** [https://anonymous.4open.science/w/casecompare_temp-4B4C/] in our Additional Response.
>
> (2) For the pipeline baseline, we implemented your suggested two-stage approach and found that it does not always improve over end-to-end modeling, confirming that visual reasoning cannot be fully captured by intermediate text alone.
>
> (3) On correlation with Interaction2Code, we reported a moderate rank correlation (~0.75) but also a large performance drop (>40 points) on IWR-Bench, showing that our benchmark introduces new, harder dimensions.
>
> These new experiments and discussions are included in the revised paper (highlighted in red). **We sincerely hope these address your concerns, could you kindly confirm whether we’ve adequately resolved them, or if further clarifications would be helpful?** Thank you for your time and invaluable insights.

---

### Official Review · Reviewer_7Tx2 · 2025-11-02

**Soundness:** 3
**Presentation:** 3
**Contribution:** 3
**Rating:** 6
**Confidence:** 3

**Summary:**

This work introduces IWR‑Bench, a benchmark that asks multimodal models to watch a video of a real website in use, along with a sheet of actual page assets, and then generate HTML/CSS/JS code that reproduces both the look and the interaction logic shown in the video. To curate IWR‑Bench, the author asks web‑dev experts to propose a large batch of candidate tasks, following which human annotators perform the workflow on the live website and record interaction videos. In parallel, they automatically collect all static assets used on the page, such as images and icons. Finally, 113 tasks are curated in IWR‑Bench. The generated websites are evaluated by replaying the workflow, and an LLM is used to score the interactive functionality and visual fidelity. The experiment results show that the SOTA LLM struggles to achieve reasonable performance. In particular, synthesizing correct interaction logic is the main bottleneck.

**Strengths:**

1. The task of synthesizing website code directly from user‑interaction video is both novel and interesting, and likely to remain challenging for multimodal LLMs in the near term.

2. Dataset curation is careful and well‑documented, with cross‑annotator verification and asset checks. Therefore, the resource should be useful to the community.

3. It's an interesting finding that LLMs mainly struggle with interactive functionality

**Weaknesses:**

1. It's understandable that building such a dataset is labor-intensive and time-consuming. But a test set with only 113 tasks might be too small-scale.

2. The evaluation metric might be too binary since a minor defect (e.g., a button a few pixels off that breaks a click) is penalized the same as a significant logic error (totally different UI element).

3. Although the proposed task is interesting, practical and non‑malicious value propositions are under‑argued. The new task would be more well-motivated if the author explained why reconstructing an existing site.

**Questions:**

See weaknesses.

---

> ### Author Response · Authors · 2025-11-19
> **Response**
>
> We thank you for the positive assessment regarding the novelty, careful curation, and the interesting findings of IWR-Bench. Detailed responses to the concerns are provided below.
>
> ### [Concern 1 (Weakness 1)] Review of the benchmark scale
>
> For clarification and to provide context, the number of tasks in IWR-Bench is comparable to established benchmarks in this domain. For instance, Interaction2Code[1] comprises 127 webpages, Design2Code[2] includes 484 screenshots ( but only 80 in Design2Code-HARD[2] ), and CC-HARD contains 128 screenshots. The design of IWR-Bench prioritizes this principle of depth and complexity. The 113 tasks are intentionally granular, comprising 1,001 distinct actions, 620 visual checkpoints, and 403 logical assertions. This granular structure is essential for rigorously evaluating the complex, stateful workflows that define this new task. Therefore, the action-level sample size of **1,001** serves as a more representative measure of the benchmark's scale and challenge.
>
> Furthermore, broad coverage is systematically ensured. Tasks are taxonomized across three orthogonal axes [Domain ( 5 major categories with 16 subcategories ), Visual Complexity ( V1/V2/V3/V4 = 11.50%/43.36%/35.40%/9.73% ), Interaction Logic ( L1/L2/L3/L4 = 7.96%/47.79%/33.63%/10.62% )] and multiple resolutions to cover a wide spectrum of real-world web scenarios. This curation strategy establishes a high-quality, high-complexity benchmark designed for comprehensive evaluation.
>
> ---
>
> ### [Concern 2 (Weakness 2)] ...evaluation metric might be too binary since a minor defect..
>
> We appreciate the opportunity to clarify the robustness of our evaluation protocol. The evaluation protocol is explicitly designed to be resilient to minor spatial or pixel-level deviations:
>
> 1. **Semantic Locators for Functionality:** The functional evaluation (agent-based) does not rely on rigid (x, y) coordinates. Elements are located using semantic descriptors (e.g., “the button with text ‘Submit’”), so minor layout shifts do not prevent the agent from interacting with the intended element. The functional correctness score is therefore unaffected by small positional changes.
>
> 2. **Tolerance in Visual Evaluation:** The Visual Fidelity Score combines OCR-based, DINO-based, and MLLM-based assessments. These metrics are robust to small pixel perturbations. A slight offset in element rendering results in negligible penalties, ensuring the score reflects overall visual logic rather than pixel-perfect alignment.
>
> Therefore, a button being a few pixels off does not cause the click to fail and is treated very differently from a genuine functional or logical error.
>
> ---
>
> ### [Concern 3 (Weakness 3)] ...why reconstructing an existing site..
>
>
> The choice of "reconstructing existing websites" is motivated by the necessity for rigorous, objective evaluation. Similar to established benchmarks like **Design2Code**[3], **WebSight**[4] and **Interaction2Code**[1] , reconstruction is utilized as a proxy task for evaluating generative capabilities. Using existing websites provides an objective and diverse **Ground Truth**, enabling precise, metric-based assessment of a model's ability to understand complex visual layouts and, crucially in IWR-Bench, **temporal interaction logic**.
>
> Without a reference ground truth, evaluation of generated interaction logic (e.g., game rules, multi-step workflows) would become inherently subjective. The reconstruction setting thus offers a more realistic diagnostic of LVLMs’ capabilities that are directly relevant to front-end automation, agentic web control, and assistive UI tools, such as rapid prototyping of functional demos.
>
>
>
> [1] Xiao, Jingyu, et al. "Interaction2Code: Benchmarking MLLM-based Interactive Webpage Code Generation from Interactive Prototyping." arXiv preprint arXiv:2411.03292 (2024).
>
> [2] Si, Chenglei, et al. "Design2code: Benchmarking multimodal code generation for automated front-end engineering." Proceedings of the 2025 Conference of the Nations of the Americas Chapter of the Association for Computational Linguistics. 2025.
>
> [3] Gui, Yi, et al. "Latcoder: Converting webpage design to code with layout-as-thought." Proceedings of the 31st ACM SIGKDD Conference on Knowledge Discovery and Data Mining V. 2. 2025.
>
> [4] Laurençon, Hugo, Léo Tronchon, and Victor Sanh. "Unlocking the conversion of web screenshots into html code with the websight dataset." arXiv preprint arXiv:2403.09029 (2024).
>
> ---
>
> Once again, we thank you for your constructive comments, which have helped us significantly improve the paper. All newly added content has been incorporated into the updated revision and highlighted in RED.

---

> > ### Author Response · Authors · 2025-11-26
> > **Short Summary of Discussion**
> >
> > Dear Reviewer 7Tx2, thank you for your constructive feedback. As the discussion period is coming to a close and it has been about a week since we posted our detailed response, we kindly ask you to review our revisions addressing your three main concerns:
> >
> > (1) **On Benchmark Scale:** We clarified that while containing 113 webpages, the benchmark includes 1,001 distinct interaction actions. This granularity ensures a depth and complexity that matches or exceeds existing static benchmarks.
> >
> > (2) **On Metric Robustness:** We explained that our semantic locators allow for minor pixel deviations, ensuring the evaluation focuses on functional logic rather than rigid alignment.
> >
> > (3) **On Motivation:** We justified "reconstruction" as a necessary proxy to establish an objective ground truth, which is critical for accurately measuring stateful interaction capabilities.
> >
> > All revisions and clarifications have been incorporated into the manuscript (highlighted in red). We would be very grateful if you could let us know whether these changes address your concerns, or if there is anything further we can clarify to help you finalize your assessment.

---

### Author Response · Authors · 2025-12-02
**Comprehensive response to AC [1/3]**

Dear ACs,

We recognize the significant added workload you've inherited due to the recent system-wide events. In an effort to assist your assessment, we offer this overview of our submission.


# 1. The Strengths Summarized by Reviewers

We sincerely thank the reviewers for their thorough and constructive feedback. We are encouraged that they reached a consensus on several key strengths of our work:

*   **Novel and Challenging Task Formulation.** Reviewers uniformly recognized the value of our proposed task. Reviewers 7Tx2 and czER highlighted it as **"novel and interesting,"** with Reviewer 7Tx2 noting it is "likely to remain challenging... in the near term." Reviewer dLFH specifically commended that our work "shifts the problem from static screenshot-to-code to interactive webpage reconstruction," filling a critical gap between static benchmarks and general video QA.

*   **High-Quality and Realistic Curation.** The rigorous data construction received strong endorsements. Reviewer vw9e praised the dataset as **"comprehensive and well-structured,"** sourced from "real-world websites ensuring realism." Reviewer 7Tx2 emphasized that the curation is "careful and well-documented," confirming the resource will be **"useful to the community."**

*   **Extensive Empirical Insights.** Reviewer vw9e noted that our evaluation across 28 models is an **"extensive empirical study"** that offers "valuable baseline results." Reviewer 7Tx2 found our analysis revealing that SOTA models "mainly struggle with interactive functionality" to be an "interesting finding," underscoring the specific capability gap our benchmark identifies.

*   **Clear Presentation and Reproducibility.** Reviewer czER rated the presentation as "excellent", commending the paper for being **"well-written"** with "detailed descriptions of data generation, evaluation metrics, and experimental setup," which ensures the work's reproducibility.

---

> ### Author Response · Authors · 2025-12-02
> **Comprehensive response to AC[2/3]**
>
> # 2. Summary of Key Concerns and Revisions
>
> We received constructive feedback from all reviewers and made substantial revisions to the manuscript to address their concerns. All changes are **highlighted in BLUE** in the revised paper. Below is a structured summary of the key issues and our responses.
>
> **Scale(Reviewers 7Tx2, dLFH)**
> - Concern: The benchmark may be considered small.
> - Response:  Given the inherent homogeneity of live webpages, we prioritized **diversity over sheer quantity**. Our task count is comparable to related benchmarks (Interaction2Code: 127 pages; Design2Code-HARD: **only 80** tasks). More importantly, the 113 tasks decompose into **1,001 action-level samples**, offering granularity that exceeds static benchmarks.
>
> **Distinction from Prior Work (Reviewer vw9e)**
> - Concern: The novelty and distinction from *Interaction2Code* needed further clarification.
> - Response: We provided a detailed **5-dimensional comparison** to show fundamental differences. Crucially, the massive **~43-point performance drop** for GPT-5 (from ~79% on *Interaction2Code* to 36.35% on IWR-Bench) empirically proves that our benchmark introduces a distinct and significantly harder set of challenges. We also added a side-by-side [visual comparison page](https://anonymous.4open.science/w/casecompare_temp-4B4C) to intuitively demonstrate these input differences.
>
> **Pipeline Baselines (Reviewer vw9e)**
> *   **Concern:** The reviewer suggested testing a two-stage pipeline (video -> description -> code).
> *   **Response:** We implemented this pipeline (results in **Appendix H**) and found no significant performance gain over end-to-end methods. This result clarifies a critical distinction: the task requires not just visual *perception*, but integrated **visual reasoning** (e.g., spatial asset grounding, layout logic).
>
>
> **Judge Reliability and Bias (Reviewer dLFH)**
> *   Concern: Using MLLM as judge could introduce bias.
> *   Response: To address this, we presented a multi-pronged validation:
>     1.  Human Verification: We reported results from a manual inspection of 100 sampled cases, which confirmed the absence of errors in logical assertion judgments.
>     2.  Quantitative Correlation: We showed that the MLLM's High-level Visual Score (HVS) has a Pearson correlation of **r=0.9806** with our objective Low-level Visual Score (LVS), demonstrating strong alignment.
>     3.  Assertion Design: We clarified that logical assertions are intentionally atomic and verifiable, minimizing ambiguity and the potential for subjective judgment.
>
> **Documentation and Transparency (Reviewers czER, dLFH)**
> *   Concern: The details on the metric, annotation process, asset statistics, video resolutions, and evaluation costs.
> *   Response: We significantly expanded the appendices to enhance transparency:
>     *   **Appendix C** documents the human alignment study validating our metric design
>     *   **Appendix I (Annotation Guidelines)** and **Appendix J (Task Sourcing Examples)** now provide the complete annotator manual and expert sourcing criteria.
>     *   **Appendix L** reports detailed asset statistics and analyzes performance correlation.
>     *   **Appendix K** provides the full video resolution distribution.
>     *   We added a cost analysis showing a full benchmark run costs between \$**1.97–12.74**, proving its accessibility.

---

> ### Author Response · Authors · 2025-12-02
> **Comprehensive response to AC[3/3]**
>
> # 3. Status of Discussion and Scores
>
> *   **Reviewer czER (Original Score: 6): This reviewer explicitly intended to raise their score but was prevented by the system shutdown.**
>   We resolved most of this reviewer's concerns. One citation issue remained, which is only an incorrect reference name and does not affect the content. The reviewer explicitly wrote: **"Once addressed, I am happy to increase my score."** We fixed it, but the system-wide suspension occurred before they could update rating. This indicates their final assessment is **more positive than the score of 6** reflects.
>
>  *   **Reviewer 7Tx2 (Original Score: 6):** Positive on novelty and curation. Before the comment system shutdown, the reviewer did not post additional comments or questions.
>
>  *   **Reviewer vw9e (Original Score: 4):**  Main concern was uniqueness vs. Interaction2Code. We provided the 5-point differentiation, the large performance gap, the pipeline baseline, and an anonymous side-by-side visualization webpage illustrating input differences. We believe the distinct contribution is now established. Before the comment system shutdown, the reviewer did not reply with further questions.
>
>  *   **Reviewer dLFH (Original Score: 4):** Concerns focused on weights and potential judge bias. We provided the human alignment study details and judge validation experiments.  We believe these responses have adequately addressed the concerns. However, before the comment system shutdown, the reviewer did not post new issues or follow-up questions.
>
>
>
> # 4. Final Claim
>
>  We fully understand the challenges and increased workload the ICLR organizing committee and the new ACs are facing. We confirm that we have never exploited any system vulnerabilities during the incident and have strictly adhered to ICLR’s Code of Ethics.
>
> Based on the constructive feedback we received during the rebuttal and the substantial revisions we have made to address all reviewer concerns, we sincerely hope this summary can help alleviate your burden in evaluating our submission. We would appreciate your consideration of our work for acceptance.

---

### Meta-Review · Area_Chair_8pyX · 2026-01-06

**Summary:**

Reviewers appreciated that the proposed task is novel and interesting, the manuscript is well-written, and a first benchmark to combine videos and ground‑truth action trajectories for web reconstruction. Reviewers also raised some concerns like the relatively smaller size of the benchmark (113 tasks) compared to other vision-language benchmarks, and differences with prior work such as Interaction2Code.

**Reviewer Concerns:**

The meta-reviewer believes that the authors provided a comprehensive rebuttal. For instance, the authors provided a detailed comparison to clarify the fundamental differences between the proposed IWR-Bench and the prior work Interaction2Code, a correlation analysis was conducted between the final scores on IWR-Bench and the full-page level metrics on Interaction2Code for nine representative models. Authors also clarified the number of tasks in IWR-Bench in the rebuttal. The 113 tasks in the benchmark are granular consisting 1,001 distinct actions, 620 visual checkpoints, and 403 logical assertions. Further, a broader coverage is systematically ensured.

**Reviewer Scores:**

One of the reviewers already seemed positive post-rebuttal. Since the main concerns such as, comparison with prior work Interaction2Code and the relatively size of the benchmark (113 tasks but 1,001 distinct actions) are comprehensively addressed in the rebuttal, the meta-reviewer believes the reviewers will have a better positive outlook if they had been able to participate fully in the discussion.

---

### Decision · Program_Chairs · 2026-01-26

Accept (Poster)